# Robust and Heavy-Tailed Mean Estimation Made Simple, via Regret Minimization

**Samuel B. Hopkins** *          **Jerry Li** †          **Fred Zhang***

## Abstract

We study the problem of estimating the mean of a distribution in high dimensions when either the samples are adversarially corrupted or the distribution is heavy-tailed. Recent developments have established efficient and (near) optimal procedures for both settings. However, the algorithms developed on each side tend to be sophisticated and do not directly transfer to the other, with many of them having ad-hoc or complicated analyses. In this paper, we provide a meta-problem and a equivalence theorem that lead to a new unified view on robust and heavy-tailed mean estimation in high dimensions. We show that the meta-problem can be solved either by a variant of the FILTER algorithm from the recent literature on robust estimation or by the quantum entropy scoring scheme (QUE), due to Dong, Hopkins and Li (NeurIPS '19). By leveraging our equivalence theorem, these results translate into simple and efficient algorithms for both robust and heavy-tailed settings. Furthermore, the QUE-based procedure has run-time that matches the fastest known algorithms on both fronts. Our analysis of FILTER is through the classic regret bound of the multiplicative weights update method. This connection allows us to avoid the technical complications in previous works and improve upon the run-time analysis of a gradient-descent-based algorithm for robust mean estimation by Cheng, Diakonikolas, Ge and Soltanolkotabi (ICML '20).

## 1 Introduction

Learning from high-dimensional data in the presence of outliers is a central task in modern statistics and machine learning. Outliers have many sources. Modern data sets can be exposed to random corruptions or even malicious tampering, as in data poison attacks. Data drawn from heavy-tailed distributions can naturally contain outlying samples—heavy-tailed data are found often in network science, biology, and beyond [22, 33, 6, 1]. Minimizing the effect of outliers on the performance of learning algorithms is therefore a key challenge for statistics and computer science.

*Robust statistics*—that is, statistics in the presence of outliers—has been studied formally since at least the 1960s, and informally since long before [27, 43]. However, handling outliers in high dimensions presents significant computational challenges. Classical robust estimators (such as the Tukey median) suffer from worst-case computational hardness, while naïve computationally-efficient algorithms (e.g., throwing out atypical-looking samples) have far-from-optimal rates of error. In the last five years, however, numerous works have developed sophisticated, efficient algorithms with optimal error rates for a variety of problems in high-dimensional robust statistics. Despite significant recent progress, many basic algorithmic questions remain unanswered, and many algorithms and rigorous approaches to analyzing them remain complex and *ad hoc*.

In this work, we revisit the most fundamental high-dimensional estimation problem, estimating the mean of a distribution from samples, in the following two basic and widely-studied robust settings. In

each case, $X_1, \ldots, X_n \in \mathbb{R}^d$ are independent samples from an unknown $d$-dimensional distribution $D$ with mean $\mu \in \mathbb{R}^d$ and (finite) covariance $\Sigma \in \mathbb{R}^{d \times d}$.

- *Robust mean estimation:* Given $Y_1, \ldots, Y_n \in \mathbb{R}^d$ such that $Y_i = X_i$ except for $\epsilon n$ choices of $i$, estimate the mean $\mu$. We interpret the $\epsilon n$ *contaminated* samples $Y_i \neq X_i$ as corruptions introduced by a malicious adversary. Naïve estimators such as the empirical mean can suffer arbitrarily-high inaccuracy as a result of these malicious samples.

- *Heavy-tailed mean estimation:* Given $X_1, \ldots, X_n$, estimate $\mu$ by an estimator $\hat{\mu}$ such that $\|\mu - \hat{\mu}\|$ is small with high probability (or equivalently, estimate $\mu$ with optimal confidence intervals). Since our only assumption about $D$ is that it has finite covariance, $D$ may have heavy tails. Standard estimators such as the empirical mean can therefore be poorly concentrated.

A significant amount of recent work in statistics and computer science has led to an array of algorithms for both problems with provably-optimal rates of error and increasingly-fast running times, both in theory and experiments [30, 14, 21, 8, 25, 11, 12, 32]. However, several questions remain, which we address in this work.

First, the relationship between heavy-tailed and robust mean estimation is still murky: while algorithms are known which simultaneously solve both problems to information-theoretic optimality [12], we lack general conditions under which algorithms for one problem also solve the other. This suggests:

> *Question 1: Is there a formal connection between robust mean estimation and heavy-tailed mean estimation which can be exploited by efficient algorithms?*

Second, iterated *sample downweighting* (or pruning) is arguably the most natural approach to statistics with outliers—indeed, the *filter*, one of the first computationally efficient algorithms for optimal robust mean estimation [14]) takes this approach—but rigorous analyses of filter-style algorithms remain *ad hoc*. Other iterative methods, such as gradient descent, suffer the same fate: they are simple-to-describe algorithms which require significant creativity to analyze [9]. We ask:

> *Question 2: Is there a simple and principled approach to rigorously analyze iterative algorithms for robust and heavy-tailed mean estimation?*

## 1.1 Our Results

Our main contribution in this work is a simple and unified treatment of iterative methods for robust and heavy-tailed mean estimation.

Addressing Question 1, we begin by distilling a simple meta-problem, which we call *spectral sample reweighing*. While several variants of spectral sample reweighing are implicit in recent algorithmic robust statistics literature, our work is the first to separate the problem from the context of robust mean estimation and show the reduction from heavy-tailed mean estimation. The goal in spectral sample reweighing is to take a dataset $\{x_i\}_{i \in [n]} \subseteq \mathbb{R}^d$, reweigh the vectors $x_i$ according to some weights $w_i \in [0, 1]$, and find a center $\nu \in \mathbb{R}^d$ such that after reweighing the maximum eigenvalue of the covariance $\sum_{i \leq n} w_i(x_i - \nu)(x_i - \nu)^\top$ is as small as possible.

**Definition 1.1** (($\alpha, \epsilon$) spectral sample reweighing, informal—see Definition 2.1). For $\epsilon \in (0, 1/2)$, let $\mathcal{W}_{n,\epsilon} = \{w \in \Delta_n : \|w\|_\infty \leq \frac{1}{(1-\epsilon)n}\}$ be the set of probability distributions on $[n]$ with bounded $\ell_\infty$ norm. Let $\alpha \geq 1$. Given $\{x_i\}_{i=1}^n$ in $\mathbb{R}^d$, an $\alpha$-approximate spectral sample reweighing algorithm returns a probability distribution $w \in \mathcal{W}_{n,3\epsilon}$ and a *spectral center* $\nu \in \mathbb{R}^d$ such that

$$\left\| \sum_{i \leq n} w_i(x_i - \nu)(x_i - \nu)^\top \right\| \leq \alpha \cdot \min_{w' \in \mathcal{W}_{n,\epsilon}, \nu' \in \mathbb{R}^d} \left\| \sum_{i \leq n} w_i'(x_i - \nu')(x_i - \nu')^\top \right\|,$$

where $\|\cdot\|$ denotes the spectral norm, or maximum eigenvalue.

Note that that spectral sample reweighing is a *worst-case* computational problem. The basic optimization task underlying spectral sample reweighing is to find weights $w \in \mathcal{W}_{n,\epsilon}$ minimizing the spectral norm of the weighted second moment of $\{x_i - \nu\}_{i \in [n]}$—an $\alpha$-approximation is allowed to

output instead $w$ in the slightly larger set $\mathcal{W}_{n,3\epsilon}$ and may only minimize the spectral norm up to a multiplicative factor $\alpha$. The parameter $\epsilon$ should be interpreted as the degree to which $w \in \mathcal{W}_{n,\epsilon}$ may deviate from the uniform distribution.

Our first result shows that robust and heavy-tailed mean estimation both reduce to spectral sample reweighing.

**Theorem 1.1** (Informal—see Theorem C.1, Theorem 4.3). *Robust and heavy-tailed mean estimation can both be solved with information-theoretically optimal error rates (up to constant factors) by algorithms which make one call to an oracle providing a constant-factor approximation to spectral sample reweighing (with $\epsilon = \epsilon_0$ a small universal constant) and run in additional time $\widetilde{O}(nd)$.*

For robust mean estimation this reduction is implicit in [14] and others (see e.g. [34]). For heavy-tailed mean estimation the reduction was not previously known: we analyze it by a simple convex duality argument (borrowing techniques from [8, 12]). Our argument gives a new equivalence between two notions of a *center* for a set of high-dimensional vectors—the spectral center considered in spectral sample reweighing and a more combinatorial notion developed by Lugosi and Mendelson in the context of heavy-tailed mean estimation [36]. We believe this equivalence is of interest in its own right—see Proposition 3.1 and Proposition 3.2.

Turning our attention to Question 2, we offer a unified approach to rigorously analyzing several well-studied algorithms by observing that each in fact instantiates a common strategy for *online convex optimization*, and hence can be analyzed by applying a standard regret bound. This leads to the following three theorems. We first demonstrate that the filter, one of the first algorithms proposed for efficient robust mean estimation [14, 35, 16, 18], can be analyzed in this framework. Specifically, we show:

**Theorem 1.2** ([14], Informal, see Theorem 2.1). *There is an algorithm,* FILTER*, based on multiplicative weights, which gives a constant-factor approximation to spectral sample reweighing for sufficiently small $\epsilon$, in time $\widetilde{O}(nd^2)$*[3].

Previous approaches to analyzing the filter required by-hand construction of potential functions to track the progress of the algorithm. Our novel strategy to prove Theorem 1.2 demystifies the analysis of the filter by applying an out-of-the-box regret bound: the result is a significantly simpler proof than in prior work. It is also general enough to capture robust mean estimation in both bounded covariance and sub-gaussian setting.

Moving on, we also analyze gradient descent, giving the following new result, which we also prove by applying an out-of-the-box regret bound. Although it gives weaker running-time bound than we prove for FILTER, the advantage is that the algorithm is vanilla gradient descent. (By comparison, the multiplicative weights algorithm of Theorem 1.2 can be viewed as a more exotic mirror-descent method.)

**Theorem 1.3** (Informal, see Theorem H.3). *There is a gradient-descent based algorithm for spectral sample reweighing which gives a constant-factor approximation to spectral sample reweighing in $O(nd^2/\epsilon^2)$ iterations and $\widetilde{O}(n^2d^3/\epsilon^2)$ time.*

Prior work analyzing gradient descent for robust mean estimation required sophisticated tools for studying non-convex iterative methods [9]. Our regret-bound strategy shows for the first time that gradient descent solves heavy-tailed mean estimation, and that it solves robust mean estimation in significantly fewer iterations than previously known (prior work shows a bound of $\widetilde{O}(n^2d^4)$ iterations in the robust mean estimation setting, where our bound gives $O(nd^2)$ iterations [9]).

Finally, we demonstrate that the nearly-linear time algorithm for robust mean estimation in [21] fits into this framework as well. Thus, this framework captures state-of-the-art algorithms for robust mean estimation.

**Theorem 1.4** ([21], Informal, see Theorem G.1). *There is an algorithm based on matrix multiplicative weights which gives a constant-factor approximation to spectral sample reweighing for sufficiently small $\epsilon$, in time $\widetilde{O}(nd\log(1/\epsilon))$.*

## 1.2 Related work

For robust mean estimation, [14, 30] give the first polynomial-time algorithm with optimal error for sub-gaussian and bounded covariance distributions. Their results have been further improved and generalized by a number of works [5, 16, 17, 19, 26, 41, 21, 15, 10, 20]. See [18] for a survey.

The first (computationally inefficient) estimator to obtain optimal confidence intervals for heavy-tailed distributions in high dimensions is given by [36]; this construction was first made algorithmic by [25], using the Sum-of-Squares method. Later works [11, 12, 32] significantly improve the run-time, with the fastest known to be $\widetilde{O}(n^2 d)$.

Analyses of the FILTER algorithm are scattered around the literature [14, 35, 16, 18]. The variant of FILTER we present here is based on a soft downweighting procedure first proposed by [40]. However, no prior work analyzes FILTER through the lens of regret minimization or points out a connection with the heavy-tailed setting.

Prior works [12, 39, 37] have proposed certain unified constructions for heavy-tailed and robust mean estimation. In particular, [12] observes a robustness guarantee of [36], originally designed for the heavy-tailed setting. However, these works do not distill a meta-problem or obtain the analysis via duality. In addition, although it matches the fastest-known running time in theory, the algorithm of [12] is based on semidefinite programming, rendering it relatively impractical. Some constructions from [39, 37] are not known to be computationally tractable.

Finally, in a concurrent and independent work, [46] also studies the spectral sample reweighing problem (in the context of robust mean estimation), and provides an analysis of filter-type algorithms based on a regret bounds. The argument of [46] relies on a technical optimization landscape analysis, which our arguments avoid. The framework of [46] can be extended to robust linear regression and covariance estimation; it is unclear our techniques extend similarly. Lastly, [46] work does not discuss the heavy-tailed setting.

## 1.3 Preliminaries and Notations

For a matrix $A$, we use $\|A\|, \|A\|_2$ to denote the spectral norm of $A$ and $\mathrm{Tr}(A)$ its trace. For a vector $v$, $\|v\|_p$ denotes the $\ell_p$ norm. We denote the all-one vector of dimension $k$ by $\mathbb{1}_k$. For PSD matrices $A, B$, we write $A \preceq B$ if $B - A$ is PSD. Density matrices refer to the set of PSD matrices with unit trace. For a weight vector $w$ such that $0 \leq w_i \leq 1$ and point set $\{x_i\}_{i=1}^n$, we define $\mu(w) = \sum_{i=1}^n w_i x_i$ and $M(w) = \Sigma_w = \sum_{i=1}^n w_i (x_i - \mu(w))(x - \mu(w))^\top$.

**Definition 1.2** (approximate top eigenvector)**.** For any PSD matrix $M$ and $c \in (0,1)$, we say that a unit vector $v$ is a $c$-approximate largest eigenvector of $M$ if $v^T M v \geq c\|M\|_2$.

**Definition 1.3** (Kullback–Leibler divergence)**.** For probability distributions $p, q$ over $[n]$, the KL divergence from $q$ to $p$ is defined as $\mathrm{KL}(p\|q) = \sum_{i=1}^n p(i) \log \frac{p(i)}{q(i)}$.

We use $\Delta_n$ to denote the set of probability distributions over $[n]$. For any $\epsilon \in (0,1)$, we define $\mathcal{W}_{n,\epsilon} = \{w \in \Delta_n : w_i \leq \frac{1}{(1-\epsilon)n}\}$; we call the set *good weights*.

## 2 The Meta-Problem and a Meta-Algorithm

We now define the meta-problem, which we call *spectral sample reweighing*, that underlies both the adversarial and heavy-tailed models. We put it as a promise problem.

**Definition 2.1** (($\alpha, \epsilon$)-spectral sample reweighing)**.** Let $\epsilon \in (0, 1/2)$. The spectral sample reweighing problem is specified by the following.

- *Input*: $n$ points $\{x_i\}_{i=1}^n$ in $\mathbb{R}^d$ and $\lambda \in \mathbb{R}$.

- *Promise*: There exists a $\nu \in \mathbb{R}^d$ and a set of good weights $w \in \mathcal{W}_{n,\epsilon}$ such that

$$\sum_{i=1}^n w_i \left(x_i - \nu\right)\left(x_i - \nu\right)^\top \preceq \lambda I. \tag{$\dagger$}$$

- *Output*: A set of good weights $w' \in \mathcal{W}_{n,3\epsilon}$ and $\nu' \in \mathbb{R}^d$ that satisfies the condition above, up to the factor of $\alpha \geq 1$:

$$\sum_{i=1}^{n} w'_i \left(x_i - \nu'\right)\left(x_i - \nu'\right)^{\top} \preceq \alpha \lambda I. \tag{2.1}$$

We will refer to the promise (†) as a *spectral centrality* assumption.

**Solving spectral sample reweighing**   Our main theorem is an efficient, constant-factor approximation algorithm for spectral sample reweighing.

**Theorem 2.1** (constant-factor approximation for spectral sample reweighing). *Let $\{x_i\}_{i=1}^{n}$ be $n$ points in $\mathbb{R}^d$ and $\epsilon \in (0, 1/10]$. Suppose there exists $\nu \in \mathbb{R}^d$ and $w \in \mathcal{W}_{n,\epsilon}$ such that*

$$\sum_{i=1}^{n} w_i \left(x_i - \nu\right)\left(x_i - \nu\right)^{\top} \preceq \lambda I$$

*for some $\lambda > 0$. Given $\{x_i\}_{i=1}^{n}, \lambda, \epsilon$ and a failure rate $\delta$, there is an algorithm that computes $w' \in \mathcal{W}_{n,3\epsilon}$ and $\nu' \in \mathbb{R}^d$ such that*

$$\sum_{i=1}^{n} w'_i \left(x_i - \nu'\right)\left(x_i - \nu'\right)^{\top} \preceq 60\lambda I,$$

*with probability at least $1 - \delta$. The algorithm runs in $O(d)$ iterations and $\widetilde{O}\left(nd^2 \log(1/\delta)\right)$ time.*

The algorithm should be seen as a variant of the FILTER algorithm [14, 40]. However, prior work did not formulate the problem quite this way or give this analysis. Instead, we will re-analyze it for spectral sample reweighing and in a different manner than previously done in the literature.

---

**Algorithm 1:** Multiplicative weights for spectral sample reweighing (Definition 2.1)

---

**Input:** A set of points $\{x_i\}_{i=1}^{n}$, an iteration count $T$, and parameter $\rho, \delta$
**Output:** A point $\nu \in \mathbb{R}^d$ and weights $w \in \mathcal{W}_{n,\epsilon}$

1 Let $w^{(1)} = \frac{1}{n}\mathbb{1}_n$ and $\eta = 1/2$.
2 **For** $t$ *from* 1 *to* $T$
3      Let $\nu^{(t)} = \sum_i w_i^{(t)} x_i$, $M^{(t)} = \sum_i w_i^{(t)}(x_i - \nu^{(t)})(x_i - \nu^{(t)})^T$.
4      Let $v^{(t)}$ be a 7/8-approximate largest eigenvector of $M^{(t)}$ (with $\|v^{(t)}\| = 1$). Fail with probability at most $\delta/T$.
5      Compute $\tau_i^{(t)} = \left\langle v^{(t)}, x_i - \nu^{(t)}\right\rangle^2$.
6      Set $w_i^{(t+1)} \leftarrow w_i^{(t)}\left(1 - \eta\tau_i^{(t)}/\rho\right)$ for each $i$.
7      Project $w^{(t+1)}$ onto the set of good weights $\mathcal{W}_{n,\epsilon}$ (under KL divergence):

$$w^{(t+1)} \leftarrow \underset{w \in \mathcal{W}_{n,\epsilon}}{\arg\min} \ \mathrm{KL}\left(w \| w^{(t)}\right)$$

8 **Return** $\nu^{(t^*)}, w^{(t^*)}$, where $t^* = \arg\min_t \|M^{(t)}\|$.

---

**Analysis via regret minimization**   We now provide a proof sketch, in the simple case of $\lambda = O(1)$, $\epsilon = 1/10$ and $v^{(t)}$ being the exact largest eigenvector. The full proof can be found in Appendix B.1.

*Proof sketch of Theorem 2.1.*   We cast the algorithm under the framework of regret minimization using multiplicative weights update (MWU). To see that, we consider $\{x_i\}_{i=1}^{n}$ as the set of actions, $w^{(t)}$ as our probability distribution over the actions at time $t$, and we receive a loss vector $\tau^{(t)}$ each round. The objective is to minimize total loss $\sum_{t=1}^{T}\langle w^{(t)}, \tau^{(t)}\rangle$. A classic regret bound of MWU [4]

shows that if $\rho = \tau_i^{(t)}$ for all $i, t$, then

$$\frac{1}{T}\sum_{t=1}^{T}\left\langle w^{(t)}, \tau^{(t)}\right\rangle \leq \frac{1}{T}(1+\eta)\sum_{t=1}^{T}\left\langle w, \tau^{(t)}\right\rangle + \frac{\rho \cdot \text{KL}(w\|w^{(1)})}{T\eta}, \tag{2.2}$$

where $\eta = 1/2$. To ensure the condition $\rho \geq \tau_i^{(t)}$ for all $i$ and $t$, observe that as $\|v^{(t)}\| = 1$, we have $\tau_i^{(t)} = \left\langle v^{(t)}, x_i - \nu^{(t)}\right\rangle^2 \leq \|x_i - \nu^{(t)}\|^2$. Also, since $\nu^{(t)}$ is a convex combination of $\{x_i\}_{i=1}^n$, we can set $\rho$ to be the squared diameter of the input data $\{x_i\}_{i=1}^n$, which, in turn, can be bounded by $O(d)$ via a simple pruning argument (Lemma B.2 and Lemma B.3).

Now note that since $v^{(t)}$ is the top eigenvector of $M^{(t)}$, for all $t$,

$$\sum_i w_i^{(t)}\tau_i^{(t)} = \sum_i w_i\left\langle v^{(t)}, x_i - \nu^{(t)}\right\rangle^2 = v^{(t)T}M^{(t)}v^{(t)} = \left\|M^{(t)}\right\|_2 \tag{2.3}$$

Let $w$ be the good weights that satisfies our centrality promise (†). By the regret bound (2.2),

$$\frac{1}{T}\sum_{t=1}^{T}\left\|M^{(t)}\right\|_2 = \frac{1}{T}\sum_{t=1}^{T}\left\langle w^{(t)}, \tau^{(t)}\right\rangle \leq (1+\eta)\frac{1}{T}\sum_{t=1}^{T}\left\langle w, \tau^{(t)}\right\rangle + \frac{\rho \cdot \text{KL}(w\|w^{(1)})}{T\eta}$$

One can check $\text{KL}(w\|w^{(1)}) \leq O(\epsilon)$ (Lemma B.8). With $\eta = 1/2$ and $\rho = O(d)$, we get

$$\frac{1}{T}\sum_{t=1}^{T}\left\|M^{(t)}\right\|_2 \leq \frac{3}{2T}\sum_{t=1}^{T}\left\langle w, \tau^{(t)}\right\rangle + \frac{O(d)}{T}. \tag{2.4}$$

The first term can be bounded by $O(\lambda)$, by expanding the definition of $\tau^{(t)}$ and applying a standard argument of second moment certificate (known as spectral signature in the literature). The second term can be bounded by $O(\lambda)$, if we set $T \gg d/\lambda$. Since $\lambda = O(1)$, this establishes the $O(d)$ iteration count. Finally, we remark that an approximate largest eigenvector can be computed in $\widetilde{O}(nd)$ time (with high probability) via power iteration, and this proves the total run-time. □

**Other algorithms for spectral sample reweighing** We note that using a matrix multiplicative update strategy, one can improve the run-time to $O(nd\log(1/\delta))$. Moreover, we provide a gradient descent-based algorithm for the problem. Our analysis leverages the same regret minimization view and improves upon that in the recent work [9]. See Appendix B.4 for further comments.

**Implications for robust mean estimation** For robust mean estimation (under bounded covariance), prior work has established that with $\Omega(d\log d/\epsilon)$ samples, a spectral center exists with $\lambda = O(1)$. Moreover, any valid spectral center achieves the optimal estimation error of $O(\sigma\sqrt{\epsilon})$. Hence, the problem of robust mean estimation can be reduced to the spectral sample reweighing problem. We work out the details in Appendix C.

## 3 Equivalent Notions of Centrality

We now give an equivalence theorem that connects heavy-tailed and robust estimation. We show that the following two (deterministic) notions of a *center* $\nu$ for points $\{x_i\}_{i=1}^k$ are essentially equivalent. We call them *spectral* and *combinatorial* center. The former is the requirement that showed up in the original formulation of the spectral sample reweighing problem (Definition 2.1), and also implicit in the prior work on robust mean estimation [16]. The latter will yield the right notion of high-dimensional median for estimating the mean of heavy tailed data, due to [36].

**Centrality notions** Recall that our meta-problem of spectral sample reweighing (Definition 2.1) requires the assumption:

$$\min_{w\in\mathcal{W}_{k,\epsilon}}\left\|\sum_{i=1}^k w_i(x_i-\nu)(x_i-\nu)^\top\right\| \leq \lambda. \tag{3.1}$$

By linearity, this can be rewritten as the following:

**Definition 3.1** (($\epsilon, \lambda$)-spectral center). A point $\nu \in \mathbb{R}^d$ is a ($\epsilon, \lambda$)-spectral center of $\{x_i\}_{i=1}^k$ if

$$\min_{w \in \mathcal{W}_{k,\epsilon}} \max_{M \succeq 0, \text{Tr}(M)=1} \sum_{i=1}^k w_i \left\langle (x_i - \nu)(x_i - \nu)^\top, M \right\rangle \leq \lambda. \qquad \text{(spectral center)}$$

Combinatorial centrality is another way of saying that the data are centered around $\nu$. It roughly claims that when we project the data onto *any* one-dimensional direction, a majority of the points will be close to $\nu$.

**Definition 3.2** (($\epsilon, \lambda$)-combinatorial center). A point $\nu$ is a ($\epsilon, \lambda$)-combinatorial center of $\{x_i\}_{i=1}^k$ if for all unit $v \in \mathbb{R}^d$.

$$\sum_{i=1}^k \mathbb{1}\left\{ \langle x_i - \nu, v \rangle \geq \sqrt{\lambda} \right\} \leq \epsilon k, \qquad \text{(combinatorial center)}$$

**Duality** It turns out that for constant $\epsilon$, these two conditions are equivalent (up to some minor gaps in constants). The proofs of the following claims can be found in Appendix D.1.

**Proposition 3.1** (spectral center $\implies$ combinatorial center). *If for some unit $v \in \mathbb{R}^d$*

$$\sum_{i=1}^k \mathbb{1}\left\{ |\langle x_i - \nu, v \rangle| \geq 10\sqrt{\lambda} \right\} \geq 0.4k, \qquad (3.2)$$

*then we have that for $\epsilon = 0.3$,*

$$\min_{w \in \mathcal{W}_{k,\epsilon}} \max_{M \succeq 0, Tr(M)=1} \sum_{i=1}^k w_i \left\langle (x_i - \nu)(x_i - \nu)^\top, M \right\rangle \geq \lambda.$$

**Proposition 3.2** (combinatorial center $\implies$ spectral center). *Let $\epsilon = 0.1$. If for some $\nu \in \mathbb{R}^d$*

$$\min_{w \in \mathcal{W}_{k,\epsilon}} \max_{M \succeq 0, Tr(M)=1} \sum_{i=1}^k w_i \left\langle (x_i - \nu)(x_i - \nu)^\top, M \right\rangle \geq \lambda$$

*then we have for some unit $v$,*

$$\sum_{i=1}^k \mathbb{1}\left\{ |\langle x_i - \nu, v \rangle| \geq 0.1\sqrt{\lambda} \right\} \geq 0.01k.$$

## 4 Estimation under Heavy-Tails

We now come to the heavy-tailed mean estimation problem and show how to solve it using the machinery developed in the last sections. The setting is very simple:

**Definition 4.1** (heavy-tailed mean estimation with optimal rates). Given $n$ random vectors $\{X_i\}_{i=1}^n$ drawn i.i.d. from a distribution $D$ over $\mathbb{R}^d$ with mean $\mu$ and (finite) covariance $\Sigma$ and a desired confidence $2^{-O(n)} \leq \delta < 1$, compute an estimate $\widehat{\mu}$ such that with probability at least $1 - \delta$,

$$\|\widehat{\mu} - \mu\| \lesssim r_\delta \overset{\text{def}}{=} \sqrt{\frac{\text{Tr}(\Sigma)}{n}} + \sqrt{\frac{\|\Sigma\| \log(1/\delta)}{n}}. \qquad (4.1)$$

We note that the error rate (4.1) is information-theoretically optimal, up to a constant. The bound is known as *sub-gaussian* error, since when $D$ is sub-gaussian, the empirical average obtains the guarantee. Moreover, in general, the estimator needs to depend on the parameter $\delta$, and the requirement that $\delta \geq 2^{-O(n)}$ is necessary [7, 13]. In the following, we will aim only at a computationally efficient, $\delta$-*dependent* construction that attains the optimal error $r_\delta$.

**Lugosi-Mendelson Estimator.** In one dimension, the well-known *median-of-means* construction, due to [38, 28, 3], provides such strong guarantee:

(i) Bucket the data into $k = \lceil 8 \log(1/\delta) \rceil$ disjoint groups and compute their means $\{Z_i\}_{i=1}^k$.

(ii) Output the median $\widehat{\mu}$ of $\{Z_i\}_{i=1}^k$.

In high dimensions, however, the question is a lot more subtle, with the correct notion of median being elusive. A long line of work culminated in the celebrated work of Lugosi and Mendelson [36]. The estimator follows the median-of-means paradigm by first bucketing the data into $k$ groups and taking the means $\{Z_i\}_{i=1}^k$. The key structural lemma of their work is that the true mean is a $(0.01, O(r_\delta^2))$-combinatorial center of the bucket means, where $r_\delta$ is the sub-gaussian error rate (4.1).

**Lemma 4.1** (Lugosi-Mendelson structural lemma [36]). *Consider the setting of heavy-tailed mean estimation (Definition 4.1). Let $\{Z_i\}_{i=1}^k$ be the $k$ bucket means with $k = \lceil 800 \log(1/\delta) \rceil$. Then with probability at least $1 - \delta$, for all unit $v \in \mathbb{R}^d$,*

$$|\langle Z_i - \mu, v \rangle| \leq 3000 \left( \sqrt{\frac{\mathrm{Tr}(\Sigma)}{n}} + \sqrt{\frac{\|\Sigma\| \log(1/\delta)}{n}} \right), \qquad (E_v)$$

*for $0.99k$ of the bucket means $\{Z_i\}_{i=1}^k$.*

This enables a natural algorithm—we can search for a point $\widehat{\mu}$ that is a $(0.01, r_\delta^2)$-combinatorial center for $\{Z_i\}_{i=1}^k$. Of course, such $\widehat{\mu}$ exists, since Lugosi-Mendelson (Lemma 4.1) showed that $\mu$ itself satisfies the condition (with probability at least $1 - \delta$). Furthermore, one can check any valid $(\epsilon, O(r_\delta^2))$-combinatorial center (Definition 3.2) $\widehat{\mu}$ with $\epsilon < 1/2$ is indeed an estimator with sub-gaussian error rate $O(r_\delta)$. We delay the proof to Appendix E.1

**Lemma 4.2** (combinatorial center is sub-gaussian). *Let $\{Z_i\}_{i=1}^k$ be defined as above and $\epsilon < 1/2$. Suppose that the condition in the Lugosi-Mendelson structural lemma (Lemma 4.1) holds. Then any $\left(\epsilon, O\left(r_\delta^2\right)\right)$-combinatorial center $\widehat{\mu}$ of $\{Z_i\}_{i=1}^k$ attains the sub-gaussian error (4.1) (up to constant).*

However, the problem of efficiently finding a combinatorial center appears difficult. If one sticks to its definition, it is required to ensure that for *all* unit vector $v$, the clustering property ($E_v$) holds. It seems that even just *certifying* this condition would naïvely take exponential time (say, by enumerating a $1/2$-net of unit sphere). Yet, we can actually resort to duality, to avoid the pain of designing a new algorithm from scratch. As we showed, a combinatorial center is just a spectral center, which our meta-algorithm can find for us. The full proof can be found in Appendix E.2.

**Theorem 4.3** (heavy-tailed mean estimation via spectral sample reweighing). *Given $\{X_i\}_{i=1}^n$ and $\delta$, any constant-factor approximation algorithm for the spectral sample reweighing problem (Definition 2.1) can be used to compute an estimate $\widehat{\mu}$ that obtains the sub-gaussian error rate for heavy-tailed mean estimation (Definition 4.1), with probability at least $1 - \delta$.*

The proof is to simply use our duality theorems, arguing that the combinatorial centrality condition from Lemma 4.1 implies spectral centrality, so we can apply the spectral sample reweighing procedure. The output is a valid spectral center, which is also a combinatorial center.

The theorem implies that the filter algorithm (Algorithm 1 combined with a simple pruning step) can be used for heavy-tailed mean estimation as well. We delay the proof to Appendix E.3.

**Corollary 4.4** (filter for heavy-tailed mean estimation). *Given $\{X_i\}_{i=1}^n$ drawn i.i.d. from a distribution with mean $\mu$ and covariance $\Sigma$ and a failure probability $2^{-O(n)} \leq \delta < 1$, there is an efficient algorithm that outputs $\widehat{\mu}$ such that with probability at least $1 - \delta$, $\|\widehat{\mu} - \mu\| \leq O(r_\delta)$.*

*Further, the algorithm is a black-box application of Algorithm 1 and runs in time $O(k^2 d^2 + nd)$.*

**Other algorithms for heavy-tailed mean estimation** One may also use the matrix multiplicative update scheme (Appendix G) to obtain an algorithm in time $\widetilde{O}(k^2 d)$. This matches the fastest-known algorithm [12, 32]. In addition, the gradient descent-based algorithm is suitable for the task as well, since it also provides a constant-approximation for spectral sample reweighing.

The formal statements and run-time analysis can be found in Appendix E.4.

## 5 Broader Impact Statement

We believe that our work is an important step on the design and analysis of efficient methods in algorithmic high-dimensional robust statistics. In particular, our unified perspective may be applicable to to other tasks in this field, so we expect it will continue to generate academic impact. Moreover, the area should be viewed as the theoretical foundation of robust and trust-worthy machine learning. Hence, we hope that our contributions will eventually have positive downstream implications for learning in sefety-critical settings.

## Footnotes

*UC Berkeley. Email: {hopkins, z0}@berkeley.edu

†Microsoft Research AI. Email: jerrl@microsoft.com

[3]We use $\widetilde{O}$ notation to hide polylogarithmic factors. Also, we remark that no efforts have been devoted to optimizing the constant breakdown point $1/10$.

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
