[Supplementary Material]


# A  More Technical Preliminaries

For vectors $u, v$, we denote the entrywise product by $u \odot v$; that is, the vector such that $w_i = u_i \cdot v_i$ for each $i$. For any symmetric matrix $A \in \mathbb{R}^{d \times d}$, let $\exp(A)$ denote the matrix exponential of $A$.

**Definition A.1** (total variation distance)**.** For probability distributions $p, q$, the total variation distance is defined as $\mathrm{TV}(p, q) = \sup_E |p(E) - q(E)| = \frac{1}{2}\|p - q\|_1$, where the supremum is over the set of measurable events.

We write $\mathcal{U}_n$ for the uniform distribution over $[n]$.

# B  Technical Details of Section 2

## B.1  Full Proof of Theorem 2.1

To prove the main theorem (Theorem 2.1) of Section 2, we first claim the following set of guarantees provided by Algorithm 1.

**Lemma B.1** (analysis of filter)**.** *Let $\epsilon \in (0, 1/10]$ and $\{x_i\}_{i=1}^n$ be $n$ points in $\mathbb{R}^d$. Suppose there exists $\nu \in \mathbb{R}^d$ and $w \in \mathcal{W}_{n,\epsilon}$ such that*

$$\sum_{i=1}^n w_i \, (x_i - \nu) \, (x_i - \nu)^\top \preceq \lambda I$$

*for some $\lambda > 0$. Then, given $\{x_i\}_{i=1}^n$, a failure rate $\delta$ and $\rho$ such that $\rho \geq \tau_i^{(t)}$ for all $i$ and $t$, Algorithm 1 finds $w' \in \mathcal{W}_{n,\epsilon}$ and $\nu' \in \mathbb{R}^d$ such that*

$$\sum_{i=1}^n w'_i \, (x_i - \nu') \, (x_i - \nu')^\top \preceq 60\lambda I, \tag{B.1}$$

*with probability at least $1 - \delta$.*

*The algorithm terminates in $T = O(\rho\epsilon/\lambda)$ iterations. Further, if $T = O(\mathrm{poly}(n, d))$, then each iteration takes $\widetilde{O}(nd \log(1/\delta))$ time.*

We first see how to prove Theorem 2.1 via Lemma B.1. Note that it requires to bound the width parameter $\rho$. To ensure the condition $\rho \geq \tau_i^{(t)}$ for all $i$ and $t$, observe that as $\|v^{(t)}\| = 1$, we have

$$\tau_i^{(t)} = \left\langle v^{(t)}, x_i - \nu^{(t)} \right\rangle^2 \leq \|x_i - \nu^{(t)}\|^2.$$

Also, since $\nu^{(t)}$ is a convex combination of $\{x_i\}_{i=1}^n$, we can set $\rho$ to be the squared diameter of the input data $\{x_i\}_{i=1}^n$. As the first step, we show that a $(1 - 2\epsilon)$ fraction of the points lie within a ball of radius $\sqrt{d\lambda/\epsilon}$ under the spectral centrality condition. Then a (folklore) pruning procedure can be used to extract such set.

**Lemma B.2** (diameter bound)**.** *Let $\{x_i\}_{i=1}^n$ be $n$ points in $\mathbb{R}^d$. Suppose there exists $\nu \in \mathbb{R}^d$ and $w \in \mathcal{W}_{n,\epsilon}$ such that $\sum_{i=1}^n w_i \, (x_i - \nu) \, (x_i - \nu)^\top \preceq \lambda I$ for some $\lambda > 0$ and $\epsilon \in (0, 1/2)$. Then there exists a ball of radius $\sqrt{d\lambda/\epsilon}$ that contains at least $r = (1 - 2\epsilon)n$ points of $\{x_i\}_{i=1}^n$.*

*Proof.* We show that there exists a ball of radius $\sqrt{d\lambda/\epsilon}$ that contains at least $(1 - 3\epsilon)n$ points. Note that the spectral centrality condition $\sum_{i=1}^n w_i \, (x_i - \nu) \, (x_i - \nu)^\top \preceq \lambda I$ implies that

$$\sum_{i=1}^n \mathrm{Tr}\left( w_i (x_i - \nu)(x_i - \nu)^\top \right) \leq d\lambda.$$

By the cyclic property of trace, we get

$$\sum_{i=1}^n w_i \|x_i - \nu\|^2 \leq d\lambda.$$

Therefore, by Markov's inequality,

$$\Pr_{i \sim w} \left( \|x_i - \nu\|^2 \geq d\lambda/\epsilon \right) \leq \epsilon, \tag{B.2}$$

where $i \sim w$ denotes $i$ drawn from the discrete distribution defined by $w$. Observe that since $\mathcal{W}_{n,\epsilon}$ is the convex hull of all uniform distributions over a subset of size $(1-\epsilon)n$, we have $\|w - \mathcal{U}_n\|_1 \leq 2\epsilon$. Thus, $\mathrm{TV}(w, \mathcal{U}_n) \leq \epsilon$. Hence, using the definition of total variation distance, (B.2) implies that

$$\Pr_{i \sim \mathcal{U}_n} \left( \|x_i - \nu\|^2 \geq d\lambda/\epsilon \right) \leq 2\epsilon, \tag{B.3}$$

as desired. $\qquad\square$

**Lemma B.3** (folklore; see [21]). *Let $\epsilon < 1/2$ and $\delta > 0$. Let $S \subset \mathbb{R}^d$ be a set of $n$ points. Assume there exists a ball $B$ of radius $r$ and a subset $S' \subseteq S$ such that $|S'| \geq (1-\epsilon)n$ and $S' \subset B$. Then there is an algorithm $\mathrm{PRUNE}(S, r, \delta)$ that runs in time $O(nd \log 1/\delta)$ and with probability $1 - \delta$ outputs a set $R \subseteq S$ so that $S' \subseteq R$, and $R$ is contained in a ball of radius $4r$.*

Using the lemmas above, we can immediately prove the main theorem.

*Proof of Theorem 2.1.* Given $S = \{x_i\}_{i=1}^n$, $\lambda$ and $\epsilon$, we first run the $\mathrm{PRUNE}(S, r, \delta/2)$ algorithm, with $r = \sqrt{d\lambda/\epsilon}$. By Lemma B.2, the spectral centrality condition (†) implies there exists a ball of radius $r$ containing at least $(1 - 2\epsilon)n$ points of $S$. Therefore, Lemma B.3 guarantees that it will return a set $R \subseteq S$ of at least $(1 - 2\epsilon)n$ points contained in a ball of radius $4r$. Hence by Lemma B.1, given $R$, $\rho = 16d\lambda/\epsilon$ and failure rate $\delta/2$, Algorithm 1 finds $w' \in \mathcal{W}_{|R|,\epsilon}$ and $\nu' \in \mathbb{R}^d$ such that

$$\sum_{i \in R} w_i' \left( x_i - \nu' \right) \left( x_i - \nu' \right)^\top \preceq 60\lambda I,$$

with probability at least $1 - \delta/2$. Let $w_i'' = w_i'$ if $i \in R$ and $w_i'' = 0$ otherwise. since $\frac{1}{(1-\epsilon)(1-2\epsilon)} \leq \frac{1}{1-3\epsilon}$ for $\epsilon < 1/3$, we have $w'' \in \mathcal{W}_{n,3\epsilon}$ Moreover, $\sum_{i=1}^n w_i'' \left( x_i - \nu' \right) \left( x_i - \nu' \right)^\top \preceq 60\lambda I$, as desired.

The overall procedure succeeds with probability at least $1 - \delta$ by a union bound, since Algorithm 1 and $\mathrm{PRUNE}$ are both set up to have a failure rate at most $\delta/2$. Now for the run-time, $\mathrm{PRUNE}(S, r, \delta)$ takes $O(nd \log(1/\delta))$ by Lemma B.3. Moreover, by Lemma B.1, Algorithm 1 runs in time $\widetilde{O}(nd \log(1/\delta) \cdot T)$ time, with $T = O(\rho\epsilon/\lambda)$ being the iteration count. Since $\rho = 16d\lambda/\epsilon$, we have $T = O(d)$, and this immediately yields the desired runtime. $\qquad\square$

**Analysis via regret minimization**  Now it remains to analyze Algorithm 1, proving Lemma B.1. We will cast the algorithm under the framework of regret minimization using multiplicative weights update (MWU). To see that, we consider $\{x_i\}_{i=1}^n$ as the set of actions, $w^{(t)}$ as our probability distribution over the actions at time $t$, and we receive a loss vector $\tau^{(t)}$ each round. The weights are updated in a standard fashion. Further, the weights must lie in the constraint set $\mathcal{W}_{n,\epsilon}$ and thus the projection step. (Note that the algorithm is implementing both the player and the adversary.) The following is a classic regret bound of MWU for the online linear optimization problem.

**Lemma B.4** (regret bound [4]). *Suppose $\rho \geq \tau_i^{(t)}$ for every $t$ and $i$. Then for any weight $w \in \mathcal{W}_{n,\epsilon}$, Algorithm 1 satisfies that*

$$\frac{1}{T} \sum_{t=1}^T \left\langle w^{(t)}, \tau^{(t)} \right\rangle \leq \frac{1}{T}(1 + \eta) \sum_{t=1}^T \left\langle w, \tau^{(t)} \right\rangle + \frac{\rho \cdot KL(w \| w^{(1)})}{T\eta}, \tag{B.4}$$

*where any choice of step size $\eta \leq 1/2$.*

In addition, we claim the following lemma and delay its proof to the appendix (Lemma B.7).

**Lemma B.5.** *Under the centrality promise (†), for any $w' \in \mathcal{W}_{n,\epsilon}$,*

$$\|\nu - \nu(w')\| \leq \frac{1}{1 - 2\epsilon} \left( \sqrt{2\lambda} + \sqrt{\epsilon \|M(w')\|} \right), \tag{B.5}$$

*where $\nu(w') = \sum_i w_i' x_i$ and $M(w') = \sum_i w_i'(x_i - \nu(w'))(x_i - \nu(w'))^\top$.*

469 This type of inequality is generally known as the *spectral signature* lemmas from the recent algorith-
470 mic robust statistics literature; see [35, 18].

471 With these technical ingredients, we are now ready to analyze the algorithm.

472 *Proof of Lemma B.1.* Notice first that since $v^{(t)}$ is a 7/8-approximate largest eigenvector of $M^{(t)} =$
473 $\sum_i w_i^{(t)}(x_i - \nu^{(t)})(x_i - \nu^{(t)})^T$, then for all $t$,

$$\sum_i w_i^{(t)} \tau_i^{(t)} = \sum_i w_i \left\langle v^{(t)}, x_i - \nu^{(t)} \right\rangle^2 = v^{(t)\top} M^{(t)} v^{(t)} \geq \frac{7}{8} \left\| M^{(t)} \right\|_2. \tag{B.6}$$

474 Let $w$ be the good weights that satisfies our centrality promise (†). Summing over the $T$ rounds and
475 applying the the regret bound (Lemma B.4), we obtain that

$$\frac{7}{8T} \sum_{t=1}^{T} \left\| M^{(t)} \right\|_2 \leq \frac{1}{T} \sum_{t=1}^{T} \left\langle w^{(t)}, \tau^{(t)} \right\rangle \leq (1+\eta) \frac{1}{T} \sum_{t=1}^{T} \left\langle w, \tau^{(t)} \right\rangle + \frac{\rho \cdot \mathrm{KL}(w \| w^{(1)})}{T\eta}.$$

476 The KL term can be bounded because $w$ and $w^{(1)}$ are both close to uniform. Indeed, it is a simple
477 calculation to verify that $\mathrm{KL}(w \| w^{(1)}) \leq 5\epsilon$, using the fact $w_i \leq 1/(1-\epsilon)n$ (Lemma B.8). Plugging
478 in $\eta = 1/2$, we get

$$\frac{7}{8T} \sum_{t=1}^{T} \left\| M^{(t)} \right\|_2 \leq \frac{3}{2T} \sum_{t=1}^{T} \left\langle w, \tau^{(t)} \right\rangle + \frac{10\epsilon\rho}{T}. \tag{B.7}$$

479 Our eventual goal is to bound this by $O(\lambda)$. Note that the second term is easy to control—just set
480 $T = \Omega(\rho\epsilon/\lambda)$, and this will determine the iteration count and thus the runtime.

481 The remaining is mostly tedious calculations to bound the first term. The reader can simply skip
482 forward to (B.13). For those interested: we proceed by expanding the first term on the right-hand
483 side,

$$\frac{3}{2T} \sum_{t=1}^{T} \langle w, \tau^{(t)} \rangle = \frac{3}{2T} \sum_{t=1}^{T} \sum_{i=1}^{n} w_i \left\langle x_i - \nu^{(t)}, v^{(t)} \right\rangle^2 \tag{B.8}$$

$$= \frac{3}{2T} \sum_{t=1}^{T} \sum_{i=1}^{n} w_i \left( \left\langle x_i - \nu, v^{(t)} \right\rangle^2 + \left\langle \nu - \nu^{(t)}, v^{(t)} \right\rangle^2 \right) \tag{B.9}$$

$$\leq \frac{3}{2}\lambda + \frac{3}{2T} \sum_{t=1}^{T} \left\langle \nu - \nu^{(t)}, v^{(t)} \right\rangle^2 \tag{B.10}$$

$$\leq \frac{3}{2}\lambda + \frac{3}{2T} \sum_{t=1}^{T} \left\| \nu - \nu^{(t)} \right\|_2^2, \tag{B.11}$$

484 where (B.8) is by the definition that $\tau_i^{(t)} = \left\langle v^{(t)}, x_i - \nu^{(t)} \right\rangle^2$, (B.9) uses the definition of $\nu^{(t)}$, (B.10)
485 follows from the spectral centrality assumption (†), and (B.11) is by the fact that $\|v^{(t)}\| = 1$. Since
486 $\nu^{(t)} = \sum_{i=1}^{n} w_i^{(t)} x_i$, we can apply Lemma B.5 to bound $\|\nu - \nu^{(t)}\|$ and it follows that

$$\frac{3}{2T} \sum_{t=1}^{T} \left\| \nu - \nu^{(t)} \right\|_2^2 \leq \frac{3}{2T} \left( \sum_{t=1}^{T} \frac{25}{2}\lambda + \frac{1}{3} \left\| M^{(t)} \right\|_2 \right),$$

487 for $\epsilon \leq 1/10$. Plugging the bound into (B.11), we obtain

$$\frac{3}{2T} \sum_{t=1}^{T} \langle w, \tau^{(t)} \rangle \leq \frac{3}{2}\lambda + \frac{3}{2T} \left( \sum_{t=1}^{T} \frac{25}{2}\lambda + \frac{1}{3} \left\| M^{(t)} \right\|_2 \right) = \frac{81}{4}\lambda + \frac{1}{2T} \sum_{t=1}^{T} \left\| M^{(t)} \right\|_2. \tag{B.12}$$

488 Finally, substituting this back into (B.7), we see that

$$\frac{7}{8T} \sum_{t=1}^{T} \left\| M^{(t)} \right\|_2 \leq \frac{81}{4}\lambda + \frac{1}{2T} \sum_{t=1}^{T} \left\| M^{(t)} \right\|_2 + \frac{10\epsilon\rho}{T}. \tag{B.13}$$

489 Now if we set $T = 10\rho\epsilon/\lambda$, then the last term is $\lambda$. Rearranging yields that $\frac{1}{T}\sum_{t=1}^{T}\left\|M^{(t)}\right\|_2 \le 60\lambda$.
490 This shows that within $T = O(\rho\epsilon/\lambda)$ iterations we have achieved our goal (B.1).

491 Now it remain to argue the cost of each iteration. For approximating the largest eigenvector, the well-
492 known power method computes a constant-approximation in $O(nd\log(1/\alpha))$ time with a failure
493 probability at most $\alpha$ [29]. We set $\alpha = \delta/T$, and an application of union bound implies that all
494 the $T$ calls to the power method jointly succeed with probability at least $1 - \delta$. This gives a total
495 run-time of $\widetilde{O}(nd\log(1/\delta))$, since $T = O(\text{poly}(n,d))$, and bounds the overall failure probability of
496 the algorithm by $\delta$. Finally, we remark that the KL projection onto $\mathcal{W}_{n,\epsilon}$ can be computed exactly in
497 $O(n)$ time, by the deterministic procedures in [24, 45]. This completes the run-time analysis. $\qquad\square$

## B.2 Proof of a Spectral Signature

499 Towards a proof of Lemma B.5, we need the following technical lemma.

500 **Lemma B.6.** *Let* $\{x_i\}_{i=1}^{n}$ *be* $n$ *points in* $\mathbb{R}^d$. *Suppose there exists* $\nu \in \mathbb{R}^d$ *and a set of good weights*
501 $w \in \mathcal{W}_{n,\epsilon}$ *such that*

$$\sum_{i=1}^{n} w_i \left(x_i - \nu\right)\left(x_i - \nu\right)^\top \preceq \lambda I. \tag{B.14}$$

502 *Then there exists a set* $G \subseteq [n]$ *of size* $(1-\epsilon)n$ *such that*

$$\frac{1}{(1-\epsilon)n}\sum_{i \in G}\left(x_i - \nu\right)\left(x_i - \nu\right)^\top \preceq \lambda I. \tag{B.15}$$

503 *Proof.* Define $w(T)$ to be the uniform distribution over a subset $T \subseteq [n]$ of data. Let $\mathcal{T}$ denote the
504 collection of all subsets of size $(1-\epsilon)n$. Observe that the set of good weights $\mathcal{W}_{n,\epsilon}$ is simply the
505 convex hull of $\{w(T) : T \in \mathcal{T}\}$. Thus, for each $i$, we can rewrite $w_i = \sum_{T \in \mathcal{T}} \alpha_T w(T)_i$ for some
506 distribution $\alpha$ over $\mathcal{T}$. Then we get that

$$\sum_{i=1}^{n} w_i \left(x_i - \nu\right)\left(x_i - \nu\right)^\top = \sum_{i=1}^{n}\sum_{T \in \mathcal{T}} \alpha_T w(T)_i \left(x_i - \nu\right)\left(x_i - \nu\right)^\top$$

$$= \sum_{T \in \mathcal{T}} \alpha_T \sum_{i=1}^{n} w(T)_i \left(x_i - \nu\right)\left(x_i - \nu\right)^\top$$

$$= \sum_{T \in \mathcal{T}} \alpha_T A_T,$$

507 where $A_T = \frac{1}{(1-\epsilon)n}\sum_{i \in T}\left(x_i - \nu\right)\left(x_i - \nu\right)^\top$. It follows that the spectral centrality condi-
508 tion (B.14) is equivalent of

$$\sum_{T \in \mathcal{T}} \alpha_T A_T \preceq \lambda I.$$

509 Therefore, there must exist a $G \in \mathcal{T}$ such that $A_G \preceq \lambda I$, as we desired in (B.15). $\qquad\square$

510 **Lemma B.7.** *Let* $\{x_i\}_{i=1}^{n}$ *be* $n$ *points in* $\mathbb{R}^d$. *Suppose there exists* $\nu \in \mathbb{R}^d$ *and a set of good weights*
511 $w \in \mathcal{W}_{n,\epsilon}$ *such that*

$$\sum_{i=1}^{n} w_i \left(x_i - \nu\right)\left(x_i - \nu\right)^\top \preceq \lambda I. \tag{B.16}$$

512 *for some* $\lambda > 0$. *Then for any* $w' \in \mathcal{W}_{n,\epsilon}$,

$$\|\nu - \nu(w')\| \le \frac{1}{1 - 2\epsilon}\left(\sqrt{\lambda} + \sqrt{2\epsilon\lambda} + \sqrt{\epsilon\|M(w')\|}\right), \tag{B.17}$$

513 *where* $\nu(w') = \sum_i w_i' x_i$ *and* $M(w') = \sum_i w_i'(x_i - \nu(w'))(x_i - \nu(w'))^\top$.

514 The lemma and its proof strategy is similar to the spectral signature lemma in robust statistics and is
515 now somewhat standard in the literature; see, *e.g.*, [21, 35].

*Proof.* First, by Lemma B.6, there exists a set $G$ of data of size $(1-\epsilon)n$ such that

$$\frac{1}{(1-\epsilon)n}\sum_{i\in G}(x_i-\nu)(x_i-\nu)^\top \preceq \lambda I. \tag{B.18}$$

Let $\mu_G = \frac{1}{(1-\epsilon)n}\sum_{i\in G}x_i$ and $B = [n]\setminus G$.

Next, to bound $\|\nu - \nu(w')\|$, we note

$$\|\nu(w')-\nu\|_2^2 = \sum_i w_i' \langle \nu(w')-\nu, x_i-\nu\rangle$$

$$= \sum_{i\in G}\frac{1}{(1-\epsilon)n}\langle \nu(w')-\nu, x_i-\nu\rangle + \sum_{i\in G}\left(w_i' - \frac{1}{(1-\epsilon)n}\right)\langle \nu(w')-\nu, x_i-\nu\rangle$$

$$+ \sum_{i\in B}w_i'\langle \nu(w')-\nu, x_i-\nu\rangle \tag{B.19}$$

We bound the three terms respectively as follows.

(i) For the first term, by Cauchy-Schwarz,

$$\sum_{i\in G}\frac{1}{(1-\epsilon)n}\langle \nu(w')-\nu, x_i-\nu\rangle = \langle \nu(w')-\nu, \mu_G-\nu\rangle \le \|\nu(w')-\nu\|\cdot\|\mu_G-\nu\|.$$

By Jensen's inequality and since $A_G \preceq \lambda I$, we have for all unit $u$,

$$\langle \mu_G-\nu, u\rangle^2 = \left\langle \frac{1}{(1-\epsilon)n}\sum_{i\in G}x_i-\nu, u\right\rangle^2 \le \frac{1}{(1-\epsilon)n}\sum_{i\in G}\langle x_i-\nu, u\rangle^2 \le \lambda.$$

Thus, $\|\mu_G-\nu\| \le \sqrt{\lambda}$.

(ii) For the second term, let $\alpha_i = w_i' - 1/(1-\epsilon)n$. Then

$$\left(\sum_{i\in G}\alpha_i\langle \nu(w')-\nu, x_i-\nu\rangle\right)^2 \le \left(\sum_{i\in G}(1-\epsilon)n\alpha_i^2\right)\cdot\sum_{i\in G}\frac{1}{(1-\epsilon)n}\langle \nu(w')-\nu, x_i-\nu\rangle^2$$

$$\le \left(\sum_{i\in G}(1-\epsilon)n\alpha_i^2\right)\cdot\lambda\cdot\|\nu(w')-\nu\|_2^2 \tag{B.20}$$

$$\le \left(\sum_{i\in G}|\alpha_i|\right)\cdot\lambda\cdot\|\nu(w')-\nu\|_2^2 \tag{B.21}$$

$$\le 2\epsilon\lambda\cdot\|\nu(w')-\nu\|_2^2, \tag{B.22}$$

where (B.20) is by the covariance bound that $A_G \preceq \lambda I$, (B.21) follows since $|(1-\epsilon)n\alpha_i| \le 1$, and (B.22) since $\sum_{i=1}^n |\alpha_i| \le \epsilon/(1-\epsilon) \le 2\epsilon$.

(iii) For the third term, we have

$$\sum_{i\in B}w_i'\langle \nu(w')-\nu, x_i-\nu\rangle = \sum_{i\in B}w_i'\langle \nu(w')-\nu, x_i-\nu(w')\rangle + \left(\sum_{i\in B}w_i\right)\|\nu(w')-\nu\|_2^2$$

$$\le \sum_{i\in B}w_i'\langle \nu(w')-\nu, x_i-\nu(w')\rangle + \epsilon\cdot\|\nu(w')-\nu\|_2^2$$

By Cauchy-Schwarz,

$$\left(\sum_{i\in B}w_i'\langle \nu(w')-\nu, x_i-\nu(w')\rangle\right)^2 \le \left(\sum_{i\in B}w_i'\right)\left(\sum_{i\in B}w_i'\cdot\langle \nu(w')-\nu, x_i-\nu(w')\rangle^2\right)$$

$$\le \epsilon\cdot\sum_{i=1}^n w_i'\langle \nu(w')-\nu, x_i-\nu(w')\rangle^2$$

$$= \epsilon\cdot(\nu(w')-\nu)^\top M(w')(\nu(w')-\nu)$$

$$\le \epsilon\cdot\|M(w')\|_2\cdot\|\nu(w')-\nu\|^2$$

528 Substituting the three bounds back into (B.19) immediately yields the result (B.17). □

## B.3 Proof of the KL Divergence Bound

530 **Lemma B.8.** *Let $p \in \mathcal{W}_{n,\epsilon}$ and $q$ be the uniform distribution over $n$ points. Then $KL(p||q) \leq 5\epsilon$.*

531 *Proof.* The lemma follows from direct calculations. By definition of KL divergence,

$$
\begin{aligned}
\mathrm{KL}(p||q) &= \sum_i p_i \log \frac{p_i}{q_i} \\
&= \sum_i p_i \log(np_i) \\
&\leq \sum_i \frac{1}{(1-\epsilon)n} \log \frac{1}{(1-\epsilon)} \\
&= \frac{1}{1-\epsilon} \log \frac{1}{1-\epsilon} \\
&\leq 5\epsilon.
\end{aligned}
$$

532 where the last inequality holds when $0 < \epsilon \leq 1/2$. □

## B.4 Further Technical Comments

534 **Faster algorithm**   Under the same assumptions, the spectral sample reweighing problem can be
535 solved in $\widetilde{O}(nd\log(1/\delta))$ time, by adapting a *matrix* multiplicative weight scheme, due to Dong,
536 Hopkins and Li [21]. The algorithm and its analysis generally follow from the proofs therein. The
537 details can be found in Appendix G.

538 As we can show in the corresponding sections, applying this procedure directly match the fastest
539 known algorithms for both robust and heavy-tailed settings.

540 **Gradient descent analysis**   As we argued, Algorithm 1 is essentially an online linear optimization
541 scheme, with the objective of minimizing $\sum_{t=1}^{T} \langle w^{(t)}, \tau^t \rangle$. It is known that the multiplicative weights
542 rule employed here can be seen an entropic mirror descent update [42]. Therefore, it is natural to
543 ask whether an additive update/gradient descent procedure would solve the problem as well. In
544 Appendix H, we provide such an analysis (Theorem H.3). More importantly, the resulting scheme
545 is equivalent to the gradient descent algorithm analyzed by [9], and our analysis improves upon
546 the iteration complexity from their work (in the concrete settings of robust mean estimation, under
547 bounded second moment and sub-gaussian distributions).

## C   Solving Robust Mean Estimation

549 We now apply Algorithm 1 for the robust mean estimation problem. We focus on the bounded second
550 moment distributions, where Algorithm 1 can be invoked in a black-box fashion. A slight variant of
551 it can be used for the sub-gaussian setting, where we achieve a more refined analysis; see Appendix
552 F.

553 The problem is formally defined below.

554 **Definition C.1** (robust mean estimation). Given a distribution $D$ over $\mathbb{R}^d$ with bounded covariance
555 and a parameter $0 \leq \epsilon < 1/2$, the adversary draws $n$ i.i.d. samples $D$, inspects the samples, then
556 removes at most $\epsilon n$ points and replaces them with arbitrary points. We call the resulting dataset
557 $\epsilon$-corrupted (by an adaptive adversary).

558 The goal is to estimate the mean of $D$ only given the $\epsilon$-corrupted set of samples.

559 Using a meta-algorithm for approximating the spectral sample reweighing problem, we will show
560 the following. In particular, using Algorithm 1 matches the run-time and statistical guarantee of the
561 original FILTER algorithm.

**Theorem C.1** (robust mean estimation via sample reweighing). *Let $D$ be a distribution over $\mathbb{R}^d$ with mean $\mu$ and covariance $\Sigma \preceq \sigma^2 I$ and $\epsilon \leq 1/10$. Given an $\epsilon$-corrupted set of $n = \Omega(d \log d/\epsilon)$ samples, there is an algorithm that runs in time $\widetilde{O}(nd^2)$ that with constant probability outputs an estimate $\widehat{\mu}$ such that $\|\widehat{\mu} - \mu\| \leq O(\sigma\sqrt{\epsilon})$.*

*Further, the algorithm is via a black-box application of Algorithm 1, which can be replaced by any algorithm solving the spectral sample reweighing problem (Definition 2.1).*

Information-theoretically, Theorem C.1 is near optimal. It is known that the sample complexity of $d \log d/\epsilon$ is tight, only up to the log factor. The estimation error $O(\sqrt{\epsilon})$ is tight up to constant factor.

Our analysis requires a set of deterministic conditions to hold for the input, which follow from Lemma A.18 of [16]. This is meant to obtain the desired spectral centrality condition and to bound the final estimation error.

**Lemma C.2** (deterministic conditions [16]). *Let $S$ be an $\epsilon$-corrupted set of $\Omega(d \log d/\epsilon)$ samples from $D$ with mean $\mu$ and covariance $\Sigma \preceq I$. With high constant probability, $S$ contains a subset $G$ of size at least $(1 - \epsilon)n$ such that*

$$\|\mu - \mu_G\| \leq O(\sqrt{\epsilon}) \tag{C.1}$$

$$\left\|\frac{1}{|G|}\sum_{i \in G}(x_i - \mu_G)(x_i - \mu_G)^\top\right\|_2 \leq O(1), \tag{C.2}$$

*where $\mu_G = \frac{1}{|G|}\sum_{i \in G} x_i$.*

We now prove the main result of this section—using the meta-algorithm to solve the robust mean estimation problem. Observe that it suffices to prove the theorem with $\sigma^2 = 1$. Without loss of generality, we can first divide every input sample by $\sigma$, execute the algorithm and then multiply the output by $\sigma$.

*Proof of Theorem C.1.* First, we check that the centrality promise (†) is satisfied. This would ensure that we are in the setting of the spectral sample reweighing problem so that the meta-algorithm applies. Assume the conditions from Lemma C.2. Then suppose we let $w_i = 1/|G|$ if $x_i \in G$ and $w_i = 0$ otherwise, so we have that $w \in \mathcal{W}_{n,\epsilon}$, and let $\nu = \mu_G$ and $\lambda = O(1)$. Observe that (C.2) is exactly the spectral centrality condition (†). Then we can apply Theorem 2.1 and obtain that the algorithm will find $\nu' \in \mathbb{R}^d$ and $w' \in \mathcal{W}_{n,3\epsilon}$ such that

$$M(w') := \sum_{i=1}^{n} w_i'(x_i - \nu')(x_i - \nu')^\top \preceq O(1) \cdot I$$

Furthermore, by definition of the algorithm, $\nu'$ is a weighted average of the points $\{x_i\}_{i=1}^{n}$; that is, $\nu' = \nu(w') = \sum_{i=1}^{n} w_i' x_i$. This allows us again to apply the spectral signature lemma. In particular, Lemma C.5 implies

$$\|\mu_G - \nu'\| \leq \frac{1}{1 - 6\epsilon}\left(\sqrt{6\epsilon\lambda} + \sqrt{3\epsilon\|M(w')\|}\right) = O\left(\sqrt{\epsilon}\right)$$

since $\lambda = O(1)$ and $\|M(w')\| = O(1)$. Finally, by triangle inequality and (C.1),

$$\|\mu - \nu'\| \leq \|\mu_G - \nu'\| + \|\mu - \mu_G\| \leq O(\sqrt{\epsilon}).$$

Therefore, the output $\nu'$ estimates the true mean up to an error of $O(\sqrt{\epsilon})$, as desired.

Finally, the run-time guarantee follows directly from the statement of Theorem 2.1, since we apply the meta-algorithm in a black-box fashion. This completes the proof. $\square$

**Other algorithms** To improve the computational efficiency, applying the same argument and using the matrix multiplicative weight algorithm (Theorem G.1), we can obtain a near linear time algorithm, which matches the fastest known algorithm for robust mean estimation [21, 10].

**Corollary C.3** (faster robust mean estimation [21]). *Let $D$ be a distribution over $\mathbb{R}^d$ with mean $\mu$ and covariance $\Sigma \preceq \sigma^2 I$ and $\epsilon$ be a sufficiently small constant. Given an $\epsilon$-corrupted set of $n = \Omega(d \log d/\epsilon)$ samples, there is a matrix multiplicative update algorithm that runs in time $\widetilde{O}(nd)$ and with constant probability computes an estimate of error $O(\sigma\sqrt{\epsilon})$.*

601 Since $\lambda = O(1)$ in the robust mean estimation problem under bounded covariance (Lemma C.2), our
602 analysis of the gradient descent algorithm (Theorem H.3) implies the following.

603 **Corollary C.4** (robust mean estimation via gradient descent). *Let $D$ be a distribution over $\mathbb{R}^d$ with*
604 *mean $\mu$ and covariance $\Sigma \preceq \sigma^2 I$ and $\epsilon$ be a sufficiently small constant. Given an $\epsilon$-corrupted set of*
605 *$n = \Omega(d \log d/\epsilon)$ samples, there is a gradient-descent based algorithm that computes an estimate of*
606 *error $O(\sigma\sqrt{\epsilon})$ with constant probability in $\widetilde{O}(n^2 d/\epsilon)$ iterations.*[2]

607 A variant of the gradient descent-based algorithm can be used for robust mean estimation in the
608 sub-gaussian setting as well; see Appendix H.2.

## C.1 Proof of the Spectral Signature

610 **Lemma C.5.** *Let $\{x_i\}_{i=1}^n$ be $n$ points. Suppose there exists a subset $G \subset [n]$ of size $(1 - \epsilon)$ such*
611 *that $\frac{1}{|G|} \sum_{i \in G} (x_i - \mu_G)(x_i - \mu_G)^\top \preceq \lambda I$ for some $\lambda > 0$, where $\mu_G = \frac{1}{|G|} \sum_{i \in G} x_i$. Then for*
612 *any $w \in \mathcal{W}_{n,\epsilon}$,*

$$\|\mu_G - \mu(w)\| \leq \frac{1}{1 - 2\epsilon}\left(\sqrt{2\epsilon\lambda} + \sqrt{\epsilon\|M(w)\|}\right). \tag{C.3}$$

613 *Proof.* The proof follows from the same argument of Lemma B.7, with $\nu = \mu_G$. Observe that the
614 first term in (B.19) becomes $\frac{1}{|G|} \sum_{i \in G} \langle \mu(w) - \nu, x_i - \nu \rangle$, which equals 0 when $\nu = \mu_G$, shaving
615 the $\sqrt{\lambda}$ term in the final bound. $\square$

# D Technical Details of Section 3

## D.1 Proof of the Duality Theorems

618 To pave way for the proofs, a key observation, first made by [8], is that the left-side of (spectral center)
619 is an SDP objective. (This is because it is simply minimizing the maximum eigenvalue of $\sum_i w_i(x_i - $
620 $\nu)(x_i - \nu)^\top$.) And strong duality allows us to swap the min and max, so

$$\min_{w \in \mathcal{W}_{k,\epsilon}} \max_M \sum_{i=1}^k w_i \left\langle (x_i - \nu)(x_i - \nu)^\top, M \right\rangle = \max_M \min_{w \in \mathcal{W}_{k,\epsilon}} \sum_{i=1}^k w_i \left\langle (x_i - \nu)(x_i - \nu)^\top, M \right\rangle,$$
(D.1)

621 where the maximization is over the set of density matrices. Using this, we prove the following two
622 propositions, showing (by contrapositives) that the two notions of centrality are equivalent. The
623 constants in the statements are chosen only to serve the purpose of heavy-tailed mean estimation,
624 and they can be tweaked easily by the same arguments.

625 We are now ready for proving the duality theorems claimed in Section 3.

626 *Proof of Proposition 3.1.* The assumption (3.2) immediately implies that

$$\sum_{i=1}^k \mathbb{1}\left\{\langle x_i - \nu, v \rangle^2 \geq 100\lambda\right\} \geq 0.4k$$

627 This means that there are (at least) $0.4k$ points in $\{x_i\}_{i=1}^k$ such that $t_i := \left\langle (x_i - \nu)(x_i - \nu)^\top, M \right\rangle \geq$
628 $100\lambda$, where $M = vv^\top$. We call them outliers.

629 Now by the SDP duality (D.1), we only need to show that for *any* feasible $w$ the objective is at
630 least $\lambda$. Observe first that for a fixed $M$, the optimal $w^*$ for the max-min objective is to put weight
631 $1/(1 - \epsilon)k$ on the $(1 - \epsilon)k$ points with the smallest $t_i$. Recall we set $\epsilon = 0.3$. Hence, by pigeonhole
632 principle, the support of $w^*$ must have an overlap of size $0.1k$ with the outliers. It follows that

$$\sum_{i=1}^k w_i^* \left\langle (x_i - \nu)(x_i - \nu)^\top, M \right\rangle \geq 0.1k \cdot \frac{1}{(1 - 0.3)k} \cdot 100\lambda \geq 10\lambda.$$

Lemma B.1 of [21].

633 Since $w^*$ is the optimal choice, this completes the proof. □

634 The other direction is a bit more involved. The key idea is to round the maximizing PSD matrix $M$
635 into a single vector $v$, via gaussian sampling, and this part of the argument is due to [12].

636 *Proof of Proposition 3.2.* Strong duality (D.1) implies that there exists PSD $M$ of unit trace such
637 that

$$\sum_{i=1}^{k} w_i \left\langle (x_i - \nu)(x_i - \nu)^\top, M \right\rangle \geq \lambda$$

638 for all $w \in \mathcal{W}_{k,\epsilon}$. As we observed, the optimal $w^*$ for a fixed $M$ would put weights on the points
639 with smallest value of $t_i = \left\langle (x_i - \nu)(x_i - \nu)^\top, M \right\rangle$. The fact that the objective is large implies that
640 there must be more than $\epsilon k = 0.1k$ points with $t_i \geq \lambda$. Let $B$ be this set of points such that $t_i \geq \lambda$.

641 It remains to demonstrate a vector $v$ such that

$$\sum_{i=1}^{k} \mathbb{1}\left\{ |\langle x_i - \nu, v \rangle| \geq 0.1\sqrt{\lambda} \right\} \geq 0.01k. \tag{D.2}$$

642 The idea is to round the PSD matrix $M$ to a single vector $v$ that achieves this inequality. The right
643 rounding method is simply *gaussian sampling*. Namely, if we draw $v_M \sim \mathcal{N}(0, M)$, then it can be
644 shown that with constant probability $v = v_M / \|v_M\|$ satisfies the property above.

For that, we apply the argument from [12]. First let $g_i = \langle x_i - \nu, v_M \rangle$ for each $i \in [k]$. Note that
$g_i$ is a mean-zero Gaussian random variable with variance $\sigma_i^2 = t_i$. A standard anti-concentration
calculation shows that for any $i \in B$, $\Pr(|g_i| \geq 0.5\sqrt{\lambda}) \geq 1/2$. Therefore, if we define

$$Y = \sum_{i=1}^{k} \mathbb{1}\left\{ |\langle x_i - \nu, v \rangle| \geq 0.5\sqrt{\lambda} \right\},$$

then by linearity of expectations we have $\mathbb{E} Y \geq 0.05k$. It follows from the Payley-Zigmund
inequality that $\Pr(Y \geq 0.01k) \geq 0.0018$. Moreover, by Borell-TIS inequality (Theorem 7.1 of [31]),
we can bound that with probability at least 0.999,

$$\|v_M\| \leq \mathbb{E}\|v_M\| + 4\sqrt{\|M\|} \leq \sqrt{\mathrm{Tr}(M)} + 4\sqrt{\mathrm{Tr}(M)} \leq 5,$$

645 since $\mathrm{Tr}(M) = 1$. Combing these facts immediately prove (D.2). □

# E    Technical Details of Section 4

## E.1    Proof of Lemma 4.2

648 The proof is via by a simple "pigeonhole + triangle inequality" argument.

649 *Proof of Lemma 4.2.* Let $v$ be the unit vector in the direction of $\mu - \widehat{\mu}$. Then since $\widehat{\mu}$ is an $(\epsilon, O(r_\delta^2))$-
650 combinatorial center with $\epsilon < 1/2$, we have $|\langle Z_i - \widehat{\mu}, v \rangle| \leq r_\delta$ for most $Z_i$. Also, $|\langle Z_i - \mu, v \rangle| \leq$
651 $O(r_\delta)$ for most $\{Z_i\}_{i=1}^{k}$ by our assumption from Lugosi-Mendelson lemma. By the pigeonhole
652 principle, there must be a $Z_j$ such that $|\langle Z_j - \widehat{\mu}, v \rangle| \leq O(r_\delta)$ and $|\langle Z_j - \mu, v \rangle| \leq O(r_\delta)$. By triangle
653 inequality,

$$\|\widehat{\mu} - \mu\| = \langle \mu - \widehat{\mu}, v \rangle \leq |\langle Z_i - \mu, v \rangle| + |\langle Z_i - \widehat{\mu}, v \rangle| \leq O(r_\delta).$$

654 as desired, and this completes the proof. □

## E.2    Proof of Theorem 4.3

656 *Proof of Theorem 4.3.* Let $\{Z_i\}_{i=1}^{k}$ be the bucket means with $k = \lceil 800 \log(1/\delta) \rceil$ and let $\lambda =$
657 $3000 r_\delta$. We assume that the true mean $\mu$ is a $(0.01, \lambda^2)$-combinatorial center of $\{Z_i\}_{i=1}^{k}$. Suppose
658 that we can obtain an $\alpha$-factor approximation the spectral sample reweighing, with the input being
659 $\{Z_i\}_{i=1}^{k}$.

- *Promise*: First let's check the spectral centrality condition holds. Since, by assumption, $\mu$ is a $(0.01, \lambda^2)$-combinatorial center of $\{Z_i\}_{i=1}^k$, we have that for all unit $v$

$$\sum_{i=1}^k \mathbb{1}\left\{|\langle x_i - \mu, v\rangle| \geq \lambda\right\} \leq 0.01k.$$

Thus, Proposition 3.2 (with $\nu = \mu$) implies that

$$\min_{w \in \mathcal{W}_{k,\epsilon}} \max_{M \succeq 0, \mathrm{Tr}(M)=1} \sum_{i=1}^k w_i \left\langle (x_i - \mu)(x_i - \mu)^T, M \right\rangle \leq 100\lambda^2,$$

where $\epsilon = 0.1$. This means that there exists $w \in \mathcal{W}_{k,\epsilon}$ such that

$$\left\| \sum_{i=1}^n w_i (x_i - \mu)(x_i - \mu)^T \right\| \leq 100\lambda^2.$$

- *Output*: Now the guarantee of an $\alpha$-factor approximation for spectral sample reweighing (Definition 2.1) is that we have $\widehat{\mu} \in \mathbb{R}^d$ and $w' \in \mathcal{W}_{k,3\epsilon}$ such that

$$\left\| \sum_{i=1}^n w_i' (x_i - \widehat{\mu})(x_i - \widehat{\mu})^T \right\| \leq 100\alpha\lambda^2.$$

It immediately follows that

$$\min_{w \in \mathcal{W}_{k,3\epsilon}} \max_{M \succeq 0, \mathrm{Tr}(M)=1} \sum_{i=1}^k w_i \left\langle (x_i - \widehat{\mu})(x_i - \widehat{\mu})^T, M \right\rangle \leq 100\alpha\lambda^2.$$

Now we can apply Proposition 3.1. Since $\alpha$ is a constant by assumption, we obtain that for all unit $v$,

$$\sum_{i=1}^k \mathbb{1}\left\{|\langle x_i - \widehat{\mu}, v\rangle| \geq C(\alpha) \cdot \lambda\right\} \leq 0.4k, \tag{E.1}$$

for some constant $C(\alpha) = O(1)$ that depends on $\alpha$. Therefore, we get that a majority of the points cluster around $\widehat{\mu}$, along any direction $v$, so it is a $(0.4, O(\lambda))$-combinatorial center. It follows from Lemma 4.2 that $\|\widehat{\mu} - \mu\| \leq O(r_\delta)$, as $\lambda = O(r_\delta^2)$.

Finally, note that the only condition of the argument is that the true mean is a combinatorial center, which occurs with probability at least $1 - \delta$, by Lemma 4.1. $\qquad\square$

We remark that the exact constants we choose in the proof are immaterial, and no efforts have been given in optimizing them.

### E.3 Proof of Corollary 4.4

*Proof of Corollary 4.4.* Given the input, we first compute the bucket means $\{Z_i\}_{i=1}^{2k}$, which takes $O(nd)$ time. Assume that the condition of the Lugosi-Mendelson structural lemma (Lemma 4.1) holds; that is, $\mu$ is a $(0.01, \lambda^2)$-combinatorial center of $\{Z_i\}_{i=1}^k$, where $\lambda = 3000r_\delta$. We use the filter algorithm (Algorithm 1) with the input being a pruned subset of $\{Z_i\}_{i=1}^k$ and apply its guarantees.

Here, we will not use the pruning step (Lemma B.3), since it requires the knowledge of $\lambda$. Instead, we first compute the coordinate-wise median-of-means $\widehat{\mu}_0$ of $\{Z_i\}_{i=k+1}^{2k}$ and the distances $d_i = \|Z_i - \widehat{\mu}_0\|$ for each $i \in [k]$. We then sort the points by $d_i$ (in descending order) and remove the top $0.01k$ points in $\{Z_i\}_{i=1}^k$ with large $d_i$. It can be shown that the remaining points has diameter at most $O(\sqrt{d}r_\delta)$; see Lemma E.1 of [32]. Let $S$ the remaining points in $\{Z_i\}_{i=1}^k$.

For the run-time, we can apply the guarantee of the filter algorithm (Lemma B.1), given the input $S$ and a failure probability $\delta/3$. Since the squared diameter is $\rho = O(dr_\delta^2)$ and $\lambda = O(r_\delta^2)$, this gives a run-time of $\widetilde{O}(k^2d^2)$, since $k = O(\log(1/\delta))$.

We now have a constant-factor approximation for the spectral sample reweighing problem. By Theorem 4.3, this gives an estimate with the sub-gaussian error (4.1). Finally, the procedure's success depends on the condition of Lugosi-Mendelson (Theorem 4.3), success of the pruning procedure, and the guarantees of constant-approximation of spectral sample reweighing (Theorem 2.1). The failure probability of each event can be bounded by $\delta/3$. Applying union bound completes the proof. □

### E.4 Further Technical Comments

**Other algorithms for heavy-tailed mean estimation** This argument also enables us to solve the heavy-tailed mean estimation problem using other approximation algorithms for the spectral sample reweighing problem. Let $\lambda = 3000r_\delta$. Recall that the argument for Theorem 4.3 shows that there is a $(0.1, O(\lambda^2))$-spectral center (which is the true mean $\mu$). Moreover, the pruning step in the proof of Corollary 4.4 allows us to bound the squared diameter of a large subset of $\{Z_i\}_{i=1}^k$ by $\rho = O(d\lambda^2)$.

This implies that the gradient descent-based algorithm that we analyze in Appendix H solves the heavy-tailed setting in $O\left(kd^2\right)$ iterations.

**Corollary E.1** (heavy-tailed mean estimation via gradient descent)**.** *Assume the setting of Corollary 4.4. A black-box application of the gradient descent-based algorithm (Algorithm 4, Appendix H) solves the heavy-tailed mean estimation problem with optimal error rate within $O(nd^2)$ iterations and $\widetilde{O}(n^2d^3)$ time.*

The quantum entropy scoring scheme (Appendix G), however, runs in $\widetilde{O}(\log(\rho/\lambda))$ number of iterations. Setting its failure probability to be $\delta/3$, we obtain the following, which matches the fastest-known algorithm for the problem [12, 32].

**Corollary E.2** (heavy-tailed mean estimation via quantum entropy scoring)**.** *Assume the setting of Corollary 4.4. A black-box application of the matrix multiplicative update algorithm (Algorithm 3, Appendix G) solves the heavy-tailed mean estimation problem with optimal error rate, in $\widetilde{O}(1)$ iterations and $\widetilde{O}(k^2d)$ total run-time.*

## F  Extension to sub-gaussian distributions

We now consider a variant of the filter algorithm (Algorithm 1) analyzed in Section 2. The difference is that instead of fixing the step size to be $\eta = 1/2$, we set it as $\epsilon$. That is, we will perform the multiplicative update less aggressively when there are few bad points. In addition, we require a stronger approximation for the largest eigenvector computation. This increases the the run-time by an $O(\text{poly}(1/\epsilon))$ factor. For technical reasons, we also ask the algorithm to stop early if the weighted covariance has been reduced to a desired value. Formally, the algorithm is described by the pseudo-code below (Algorithm 2).

---

**Algorithm 2:** Multiplicative weights for sub-gaussian robust mean estimation

**Input:** A set of points $\{x_i\}_{i=1}^n$, an iteration count $T$, and parameter $\rho, \delta$
**Output:** A set of weights $w \in \mathcal{W}_{n,\epsilon}$.

1 Let $w^{(1)} = \frac{1}{n}\mathbb{1}_n$.
2 **For** $t$ *from* $1$ *to* $T$
3      Let $\nu^{(t)} = \sum_i w_i^{(t)} x_i$, $M^{(t)} = \sum_i w_i^{(t)}(x_i - \nu^{(t)})(x_i - \nu^{(t)})^T$.
4      Let $v^{(t)}$ be a $(1 - \epsilon^2)$-approximate largest eigenvector of $M^{(t)}$ (with $\|v^{(t)}\| = 1$). Fail with probability at most $\delta/T$.
5      **If** $\lambda^{(t)} = v^{(t)\top} M^{(t)} v^{(t)} \leq 1$, **return** $w^{(t)}$.
6      Compute $\tau_i^{(t)} = \left\langle v^{(t)}, x_i - \nu^{(t)} \right\rangle^2$.
7      Set $w_i^{(t+1)} \leftarrow w_i^{(t)} \left(1 - \epsilon\tau_i^{(t)}/\rho\right)$ for each $i$.
8      Project $w^{(t+1)}$ onto the set of good weights $\mathcal{W}_{n,\epsilon}$ (under KL divergence).
9 **Return** $w^{(t^*)}$, where $t^* = \arg\min_t \|M^{(t)}\|$.

---

First, we need a stronger spectral signature lemma.

**Lemma F.1** ([21])**.** *Let $S = \{x_i\}_{i=1}^n$ be an $\epsilon$-corrupted set of $n$ samples from a sub-gaussian distribution over $\mathbb{R}^d$, with mean $\mu$ and identity covariance. Suppose $n \geq \widetilde{\Omega}(d/\epsilon^2)$. If $\|M(w)\| \leq 1 + \lambda$, for some $\lambda \geq 0$, then for any $w \in \mathcal{W}_{n,2\epsilon}$,*

$$\|\mu - \mu(w)\| \leq \frac{1}{1 - \epsilon}\left(\sqrt{\epsilon\lambda} + C\epsilon\sqrt{\log(1/\epsilon)}\right),$$

*for some universal constant $C > 0$.*

Moreover, we assume that for all $w \in \mathcal{W}_{n,2\epsilon}$ we have

$$\left\|\sum_{i \in G} w_i(x_i - \mu)(x_i - \mu)^\top - I\right\| \leq \lambda = O(\epsilon \log(1/\epsilon)). \tag{F.1}$$

This condition holds with high probability over the draws of samples [14].

**Lemma F.2** (analysis of sub-gaussian filter)**.** *Let $\epsilon$ be a sufficiently small constant and $\{x_i\}_{i=1}^n$ be $n$ points in $\mathbb{R}^d$. Assume the following (deterministic) conditions hold.*

*(i) There exists $\nu \in \mathbb{R}^d$ and $w \in \mathcal{W}_{n,\epsilon}$ such that*

$$\left\|\sum_{i=1}^n w_i (x_i - \nu)(x_i - \nu)^\top\right\| \leq 1 + O(\epsilon \log(1/\epsilon)). \tag{F.2}$$

*(ii) If $\|M(w)\| \leq 1 + \lambda$, for some $\lambda \geq 0$, then for any $w \in \mathcal{W}_{n,\epsilon}$,*

$$\|\nu - \mu(w)\| \leq \frac{1}{1 - \epsilon}\left(\sqrt{\epsilon\lambda} + C\epsilon\sqrt{\log(1/\epsilon)}\right), \tag{F.3}$$

*Then, given $\{x_i\}_{i=1}^n$, a failure rate $\delta$ and $\rho$ such that $\rho \geq \tau_i^{(t)}$ for all $i$ and $t$, Algorithm 2 finds $w' \in \mathcal{W}_{n,\epsilon}$ such that*

$$\|M(w')\| \leq 1 + O(\epsilon \log(1/\epsilon)), \tag{F.4}$$

*with probability at least $1 - \delta$.*

*The algorithm terminates in $T = O(\rho/\epsilon)$ iterations. Further, if $T = O(poly(n,d))$, then each iteration takes $\widetilde{O}(nd \log(1/\delta)/\epsilon^2)$ time.*

*Proof of Lemma F.2.* If the algorithm gets stopped early (at Line 5), then it means that

$$\|M^{(t)}\| \leq \lambda^{(t)}/(1 - \epsilon^2) \leq 1/(1 - \epsilon^2) \leq 1 + O(\epsilon^2),$$

since $v^{(t)}$ is a $(1 - \epsilon^2)$ approximate largest eigenvector of $M^{(t)}$. Hence, in this case, we immediately achieves the goal (F.4).

Now assume the algorithm did not stop early and so $\|M^{(t)}\| > 1$ for all $t$. Then we have

$$\sum_i w_i^{(t)}\tau_i^{(t)} = \sum_i w_i^{(t)}\left\langle v^{(t)}, x_i - \nu^{(t)}\right\rangle^2 = v^{(t)\top}M^{(t)}v^{(t)} \geq (1 - \epsilon^2)\left\|M^{(t)}\right\|_2, \tag{F.5}$$

for all $t$. Since the step size $\epsilon < 1/2$ and $\rho \geq \tau_i^{(t)}$ for all $i, t$ by assumption, we can apply the regret bound of MWU (Lemma B.4) and conclude that for $w$ that satifies assumption (F.2),

$$\frac{1 - \epsilon^2}{T}\sum_{t=1}^T\left\|M^{(t)}\right\|_2 \leq \frac{1}{T}\sum_{t=1}^T\left\langle w^{(t)}, \tau^{(t)}\right\rangle \leq (1 + \epsilon)\frac{1}{T}\sum_{t=1}^T\left\langle w, \tau^{(t)}\right\rangle + \frac{\rho \cdot \text{KL}(w\|w^{(1)})}{T\epsilon}. \tag{F.6}$$

We now focus on bounding $\frac{1}{T}\sum_{t=1}^T\left\langle w, \tau^{(t)}\right\rangle$.

**Claim F.3.** *In the setting above, we have*

$$\frac{1}{T}\sum_{t=1}^T\left\langle w, \tau^{(t)}\right\rangle \leq 1 + O(\epsilon \log(1/\epsilon)) + \frac{2\epsilon}{(1 - \epsilon)^2}\frac{1}{T}\sum_{t=1}^T\left\|M^{(t)}\right\| - \frac{2\epsilon}{(1 - \epsilon)^2}$$

744 *Proof.* Note that

$$\frac{1}{T}\sum_{t=1}^{T}\left\langle w,\tau^{(t)}\right\rangle = \frac{1}{T}\sum_{t=1}^{T}\sum_{i=1}^{n}w_i\left\langle x_i-\nu^{(t)},v^{(t)}\right\rangle$$

$$= \frac{1}{T}\sum_{t=1}^{T}\sum_{i=1}^{n}w_i\left(\left\langle x_i-\nu,v^{(t)}\right\rangle^2+\left\langle\nu-\nu^{(t)},v^{(t)}\right\rangle^2\right)$$

$$\leq 1+O\left(\epsilon\log(1/\epsilon)\right)+\frac{1}{T}\sum_{t=1}^{T}\left\langle\nu-\nu^{(t)},v^{(t)}\right\rangle^2 \tag{F.7}$$

$$\leq 1+O\left(\epsilon\log(1/\epsilon)\right)+\frac{1}{T}\sum_{t=1}^{T}\left\|\nu-\nu^{(t)}\right\|_2^2, \tag{F.8}$$

745 where (F.7) follows from the assumption (F.4). Now we apply assumption (F.3) to bound $\left\|\nu-\nu^{(t)}\right\|_2^2$.
746 Since we may assume $\|M^{(t)}\|\geq 1$ by the early stopping of Line 5, we have

$$\left\|\nu-\nu^{(t)}\right\|^2 \leq \frac{2}{(1-\epsilon)^2}\left(\epsilon\left(\left\|M^{(t)}\right\|-1\right)+C^2\epsilon^2\log(1/\epsilon)\right)$$

$$= \frac{2\epsilon}{(1-\epsilon)^2}\left\|M^{(t)}\right\|-\frac{2\epsilon}{(1-\epsilon)^2}+O(\epsilon^2\log(1/\epsilon)).$$

747 Substituting the bound back into (F.8) completes the proof. □

748 Using Claim F.3, the KL bound (Lemma B.8) and (F.6), we have

$$\frac{1-\epsilon^2}{T}\sum_{t=1}^{T}\left\|M^{(t)}\right\|_2 \leq \frac{2(1+\epsilon)\epsilon}{(1-\epsilon)^2}\frac{1}{T}\sum_{t=1}^{T}\left\|M^{(t)}\right\|+1-\frac{2(1+\epsilon)\epsilon}{(1-\epsilon)^2}+O(\epsilon\log(1/\epsilon))+\frac{5\rho}{T}.$$

749 For sufficiently small $\epsilon$, we rearrange and divide through to obtain

$$\frac{1}{T}\sum_{t=1}^{T}\left\|M^{(t)}\right\|_2 \leq 1+O(\epsilon\log(1/\epsilon))+O(\epsilon)+\frac{O(\rho)}{T}.$$

750 Setting $T=O(\rho/\epsilon)$ completes the correctness proof. Finally, the per-iteration cost follows from the
751 run-time of using power method to approximate the largest eigenvector. □

752 Using the lemma we can prove our main theorem.

753 **Theorem F.4** (sub-gaussian robust mean estimation, [14])**.** *Let $S=\{x_i\}_{i=1}^{n}$ be an $\epsilon$-corrupted set of*
754 *$n$ samples from a sub-gaussian distribution over $\mathbb{R}^d$, with mean $\mu$ and identity covariance. Suppose*
755 *$n\geq\widetilde{\Omega}(d/\epsilon^2)$. Given $S$, there is an algorithm that outputs $\widehat{\mu}$ such that $\|\widehat{\mu}-\mu\|\leq O\left(\epsilon\log\left(1/\epsilon\right)\right)$*
756 *with high constant probability. The algorithm runs in time $\widetilde{O}\left(nd^2/\epsilon^3\right)$*

757 *Proof.* Let $\delta=0.01$. We apply Algorithm 2 with a simple pruning procedure as a preprocessing. By
758 standard concentration of sub-gaussian random vectors, with high constant probability, $\|x_i-\mu\|\leq$
759 $r=O(\sqrt{d\log n})$ for all $i\in G$. Hence, we apply PRUNE$(S,r,\delta)$, and by Lemma B.3 it guarantees
760 to terminate in $O(nd)$ time and removes at most $\epsilon n$ (bad) points.

761 We feed the remaining (at least) $(1-\epsilon)n$ points $R\supseteq G$ into Algorithm 2 with $\rho=r^2$. Notice that
762 $\frac{1}{(1-\epsilon)(1-\epsilon)}\leq\frac{1}{1-2\epsilon}$ for $\epsilon\leq 1/2$, so assumptions (i)-(ii) of Lemma F.2 are satisfied by the claim of
763 (F.1) and Lemma F.1, respectively.

764 It then follows from Lemma F.2 that Algorithm 2 outputs $w'\in\mathcal{W}_{|R|,\epsilon}$ such that

$$\left\|\sum_{i\in R}(x_i-\mu(w'))(x_i-\mu(w'))^\top\right\|\leq 1+O\left(\epsilon\log(1/\epsilon)\right),$$

765 where $\mu(w')=\sum_{i\in R}w_i'x_i$. Let $w_i''=w_i'$ if $i\in R$ and $w_i''=0$ otherwise. We obtain $w''\in$
766 $\mathcal{W}_{n,2\epsilon}$ such that $\|M(w'')\|\leq 1+O\left(\epsilon\log\left(1/\epsilon\right)\right)$. Applying the spectral signature (Lemma F.1)
767 proves that $\mu(w'')$ attains the desired estimation error. Moreover, the run-time simply follows from
768 Lemma F.2. □

# G   Sampling reweighing via Matrix Multiplicative Update

We now show that the spectral sample reweighing problem (Definition 2.1) can be solved in near linear time via a matrix multiplicative update scheme from the recent work of [21]. Our analysis will closely resemble the arguments therein.

**Theorem G.1.** *Let $\{x_i\}_{i=1}^n$ be $n$ points in $\mathbb{R}^d$. Suppose there exists $\nu \in \mathbb{R}^d$ and $w \in \mathcal{W}_{n,2\epsilon}$ such that $\sum_{i=1}^n w_i (x_i - \nu)(x_i - \nu)^\top \preceq \lambda I$ for some $\lambda > 0$ and a sufficiently small $\epsilon$. Then, given $\{x_i\}_{i=1}^n, \lambda$, the squared diameter $\rho$ of the points and a failure rate $\delta$, there is a matrix multiplicative weights-based algorithm (Algorithm 3) that, with probability at least $1 - \delta$, finds $w' \in \mathcal{W}_{n,\epsilon}$ and $\nu' \in \mathbb{R}^d$ such that*

$$\sum_{i=1}^n w'_i (x_i - \nu')(x_i - \nu')^\top \preceq O(\lambda)I.$$

*Further, the algorithm terminates in $O(\log(\rho/\lambda))$ iterations, where $\rho$ is the squared diameter of the input points $\{x_i\}_{i=1}^n$, and each iteration can be implemented in $\widetilde{O}(nd \log(1/\delta))$ time.*

**Remark G.1.** In the following, we will consider an idealized version of the algorithm and omit the detail of implementing the numerical linear algebra primitives in $\widetilde{O}(nd \log(1/\delta))$ time each iteration. The exact details can be found in [21].

The algorithm is based on the matrix multiplicative weights update. For a sequence of PSD matrices $M_1 \succeq M_2 \succeq \cdots \succeq M_{t-1}$, we will apply the matrix multiplicative weight (MMW) update, given by

$$\mathsf{MMW}(M_0, M_1, \cdots, M_{t-1}) = \exp\left(\frac{1}{\|M_0\|_2} \sum_{k=1}^{t-1} M_k\right) \Big/ \operatorname{tr} \exp\left(\frac{1}{\|M_0\|_2} \sum_{k=1}^{t-1} M_k\right). \quad \text{(G.1)}$$

For technical reasons, we will not maintain a set of weights that is a probability distribution throughout. Instead, recall by Lemma B.6 that there exists a subset $G$ of size $(1 - \epsilon)n$ such that $A_G \preceq \lambda I$, where $A_G = \frac{1}{(1-\epsilon)n} \sum_{i \in G} (x_i - \nu)(x_i - \nu)^\top$. Thus, our new notion of a good set of weights is that starting from the uniform distribution over $n$ points, more weights are removed from $[n] \setminus G$ than from $G$. Let $w_G, w_B$ denote the restriction of $G$ to the indices of vector $w$ and $B = [n] \setminus G$.

**Definition G.1** (mostly-good weight vector). Given $\{x_i\}_{i=1}^n$ that satisfy the spectral centrality condition (†), let $G$ be a subset of size $(1 - \epsilon)n$ such that $A_G \preceq \lambda I$. The set of mostly-good weight vectors is

$$\mathcal{C}_{n,\epsilon} = \left\{ w \in \mathbb{R}^n : 0 \le w_i \le \frac{1}{n} \quad \text{and} \quad \left| \frac{1}{n} \mathbb{1}_{|G|} - w_G \right| \le \left| \frac{1}{n} \mathbb{1}_{|B|} - w_B \right| \right\}$$

A crucial subroutine we use is a deterministic down-weighting scheme, directly from [21], that maintains the mostly-good property of the input weights.

**Lemma G.2** (1D Filter [21]). *Let $\eta \in (0, 1/2)$, let $b \ge 2\eta$, and let $w_1, \ldots, w_m$ and $\tau_1, \ldots, \tau_m$ be non-negative numbers so that $\sum_{i=1}^m w_i \le 1$. Let $\tau_{\max} = \max_{i \in [m]} \tau_i$. Suppose there exist two disjoint sets $G, B$ so that $G \cup B = [m]$, and moreover,*

$$\sum_{i \in G} w_i \tau_i \le \eta \sigma \,, \quad \text{where} \quad \sigma = \sum_{i=1}^n w_i \tau_i \,.$$

*Then* $1\mathrm{DFILTER}(w, \tau, b)$ *runs in time* $O((1 + \log \frac{\tau_{\max}}{b\sigma})m)$ *and outputs* $0 \le w' \le w$ *so that:*

- *more weight is removed from $B$ than from $G$: $\sum_{i \in G} w_i - w'_i \le \sum_{i \in B} w_i - w'_i$, and*

- *the weighted sum of $\tau_i$ is decreased: $\sum_{i=1}^m w'_i \tau_i \le b\sigma$.*

The algorithm is formally described in Algorithm 3. Throughout let $M^{(s)} = M(w^{(s)})$ and $M_t^{(s)} = M(w_t^{(s)})$, where $M(w) = \sum_{i=1}^n w_i(x_i - \mu(w))(x_i - \mu(w))^\top$. The procedure runs by epochs, where each epoch $s$ reduces the largest eigenvalue of $M^{(s)}$ by a constant factor. We will show that the inner loop achieves the reduction within $O(\log d)$ iterations while maintaining the invariant that the weights are mostly-good (Definition G.1).

---

**Algorithm 3:** Matrix multiplicative update for spectral sample reweighing (Definition 2.1)

**Input:** A set of points $x_1, \ldots, x_n$, $\lambda, \rho$ and a failure rate $\delta$
**Output:** A point $\nu' \in \mathbb{R}^d$ and weights $w' \in \mathcal{W}_{n,\epsilon}$ that satisfy (2.1) up to a constant factor.

**1** Let $w^{(0)} = \frac{1}{n}(1, 1, \cdots, 1)$.
**2** **For** $s$ *from* $0$ *to* $O(\log \rho)$
**3**     Compute $\lambda^{(s)} = \|M^{(s)}\|$.
**4**     **If** $\lambda^{(s)} \leq 300\lambda$
**5**       **Return** $w^{(s)}/\|w^{(s)}\|_1, \mu(w^{(s)})$.
**6**     **For** $t$ *from* $0$ *to* $O(\log d)$
**7**       Compute $\lambda_t^{(s)} = \|M_t^{(s)}\|$ and terminate epoch **if** $\lambda_t^{(s)} \leq \frac{2}{3}\lambda_0^{(s)}$.
**8**       Compute $U_t^{(s)} = \mathsf{MMW}(M_1^{(s)}, M_2^{(s)}, \cdots, M_{t-1}^{(s)})$.
**9**       Compute

$$\tau_{t,i}^{(s)} = \left(x_i - \mu\left(w_t^{(s)}\right)\right)^\top U_t^{(s)} \left(x_i - \mu\left(w_t^{(s)}\right)\right) \tag{G.2}$$

**10**       Let $w_{t+1}^{(s)} = w_t^{(s)}$ **if** $\sum_i w_{t,i}^{(s)}\tau_{t,i}^{(s)} \leq \frac{1}{4}\lambda_1^{(s)}$; **otherwise** $w_{t+1}^{(s)} = 1\mathsf{DFILTER}(w_t^{(s)}, \tau_t^{(s)})$.
**11**     Let $w^{(s+1)} = w_t^{(s)}$.

---

To start with the analysis, we first establish certain invariants of the algorithm. This requires the following lemma. The proof follows from exactly the same argument for Lemma B.7, which we omit for the sake of brevity.

**Lemma G.3.** *Let $\{x_i\}_{i=1}^n$ be $n$ points in $\mathbb{R}^d$. Suppose there exists $\nu \in \mathbb{R}^d$ and a mostly-good weight vector $w \in \mathcal{C}_{n,\epsilon}$ such that $\sum_{i=1}^n w_i (x_i - \nu)(x_i - \nu)^\top \preceq \lambda I$, for some $\lambda > 0$. Then for any $w' \in \mathcal{C}_{n,\epsilon}$,*

$$\|\nu - \nu(w')\| \leq \frac{1}{1 - 2\epsilon}\left(2\sqrt{\lambda} + \sqrt{\epsilon\|M(w')\|}\right), \tag{G.3}$$

*where $\nu(w') = \sum_i w_i' x_i$ and $M(w') = \sum_i w_i'(x_i - \nu(w'))(x_i - \nu(w'))^\top$.*

Using this, we establish a key lemma of the inner loop of the algorithm.

**Lemma G.4.** *Let $w \in \mathcal{C}_{n,\epsilon}$ be such that $\beta = \|M(w)\|_2 \geq 300\lambda$ and $U$ be a density matrix. Let $\tau_i = (x_i - \mu(w))^\top U (x_i - \mu(w))$. If $\beta \geq \frac{1}{4}\beta$ and $w' = 1\mathsf{DFILTER}(w, \tau, 1/4)$, then we have $w' \in \mathcal{C}_{n,\epsilon}$ and $\langle M(w'), U \rangle \leq \frac{1}{4}\langle M(w), \bar{U} \rangle$.*

*Proof.* Let $\widetilde{w}_i = 1/n$ if $i \in G$ and $\widetilde{w}_i = 0$ otherwise. Let $\mu(\widetilde{w}) = \sum_i \widetilde{w}_i x_i$. Then for any unit vector $u$, we have

$$\langle \mu(\widetilde{w}) - \nu, u \rangle^2 \leq \left\langle \frac{1}{(1-\epsilon)n}\sum_{i \in G} x_i - \nu, u \right\rangle^2 \leq \frac{1}{(1-\epsilon)n}\sum_{i \in G}\langle x_i - \nu, u \rangle^2 \leq \lambda.$$

814  Thus, $\|\mu(\widetilde{w}) - \nu\|_2^2 \le \lambda$. Expanding the definition of $\tau_i$, we get

$$\sum_{i \in G} w_i \tau_i = \left\langle \sum_{i \in G} w_i \left(X_i - \mu(w)\right)\left(X_i - \mu(w)\right)^\top, U \right\rangle$$

$$\le \left\langle \sum_{i=1}^n \widetilde{w}_i \left(X_i - \mu(w)\right)\left(X_i - \mu(w)\right)^\top, U \right\rangle$$

$$= \left\langle \sum_{i=1}^n \widetilde{w}_i \left(X_i - \mu(\widetilde{w})\right)\left(X_i - \mu(\widetilde{w})\right)^\top, U \right\rangle + \|\widetilde{w}\|_1 \cdot (\mu(\widetilde{w}) - \mu(w))^\top U(\mu(\widetilde{w}) - \mu(w))$$

$$\le \langle M(\widetilde{w}), U \rangle + (1-\epsilon)\|\mu(\widetilde{w}) - \mu(w)\|_2^2$$

$$\le \lambda + 2\|\mu(\widetilde{w}) - \nu\|_2^2 + 2\|\mu(w) - \nu\|_2^2 \tag{G.4}$$

$$\le 3\lambda + (5\lambda + 2\epsilon\|M(w)\|) \tag{G.5}$$

$$\le \frac{1}{30}\|M(w)\| = \frac{1}{30}\sum_i w_i \tau_i, \tag{G.6}$$

815  where (G.4) follows since $M(\widetilde{w}) \preceq A_G \preceq \lambda I$, (G.5) follows from Lemma G.4, and (G.6) uses our
816  assumption that $\|M(w)\| \ge 300\lambda$ and the definition of $\tau_i$. This allows us to apply the guarantee of
817  the 1D filter procedure (Lemma G.2) and get that

$$\langle M(w'), U \rangle = \left\langle \sum_{i=1}^n w_i' \left(X_i - \mu(w)\right)\left(X_i - \mu(w)\right), U \right\rangle = \sum_{i=1}^n w_i' \tau_i \le \frac{1}{4}\sum_{i=1}^n w_i \tau_i = \frac{1}{4}\langle M(w), U \rangle.$$

818  Furthermore, $w' \in \mathcal{C}_{n,\epsilon}$.This completes the proof. □

819  We are now ready to prove the main theorem of this section.

820  *Proof of Theorem G.1.* Consider a fixed epoch and drop the super script for simplicity of notation.
821  It is not hard to observe that $M(w_{t+1}) \preceq M(w_t)$ (see Lemma 3.4 [21]). Let $\alpha = 1/\|M(w_0)\|$. A
822  regret bound of matrix multiplicative weights [2] implies that

$$\left\|\sum_{t=0}^{T-1} M(w_{t+1})\right\|_2 \le \sum_{t=0}^{T-1} \langle M(w_{t+1}), U_t \rangle + \alpha \sum_{t=0}^{T-1} \langle U_t, M(w_{t+1}) \rangle \|M(w_{t+1})\|_2 + \frac{\log d}{\alpha}$$

$$\le 2\sum_{t=0}^{T-1} \langle M(w_{t+1}), U_t \rangle + \|M(w_0)\|_2 \cdot \log d$$

823  Now by definition of Line 10, we have $\langle M(w_{t+1}), U_t \rangle \le \frac{1}{4}\|M(w_0)\|_2$. Hence,

$$T\|M(w_T)\|_2 \le \left\|\sum_{t=0}^{T-1} M(w_t)\right\|_2 \le T \cdot \frac{1}{2}\|M(w_0)\|_2 + \|M(w_0)\|_2 \cdot \log d.$$

824  Setting $T \gg \log d$ shows that the inner loop terminates in $O(\log d)$ iterations and reduces the largest
825  eigenvalue of the covariance by, say, $4/5$.

826  Finally, to bound the number of epochs, we simply note that $\|M^{(0)}\| \le \rho$. Therefore, $O(\log(\rho/\lambda))$
827  epochs suffice drive the largest eigenvalue of $\|M^{(s)}\|$ down to $O(\lambda)$, since it is reduced geometrically
828  each epoch. □

# H   Sample reweighing via Online Gradient Descent

## H.1   Regret analysis of gradient descent

831  We now consider a gradient updated-based algorithm for solving the spectral sample reweighing
832  problem (Definition 2.1). The analysis will be through the classic regret guarantee of online gradient
833  descent for convex optimization [46]. Though the resulting run-time is higher than the MWU scheme

834 we analyzed in Section 2, it nonetheless betters the recent work of [9], where essentially the same
835 gradient descent-based algorithm is studied.

836 We will leverage the following regret guarantee of online gradient descent; the definition of the
837 algorithm in the general setting can be found in standard text [23].

838 **Lemma H.1** (Theorem 3.1 [23], originally due to [46]). *Let $f_t : \mathcal{K} \to \mathbb{R}$ be the convex cost function*
839 *revealed at iteration t, where $\mathcal{K}$ is a convex feasible set. Suppose $f_t$ is L-Lipschitz (in $\ell_2$ norm) and*
840 $\|x_0 - x^*\|_2 \leq R$ *for some* $x^* \in \arg\min_{x \in \mathcal{K}} \sum_t f_t(x)$. *The online gradient descent algorithm with*
841 *step sizes* $\eta_t = \frac{R}{L\sqrt{t}}$ *achieves*

$$\sum_{t=1}^{T} f_t(x_t) - \min_{x \in \mathcal{K}} \sum_{t=1}^{T} f_t(x) \leq \frac{3}{2} L R \sqrt{T}. \tag{H.1}$$

842 Our algorithm implicitly defines the cost functions $f_t(w) = \langle w, \tau^{(t)} \rangle$, where the feasible set is
843 $\mathcal{W}_{n,\epsilon}$, and implements the online gradient descent algorithm for the linear objective. Note that
844 $\nabla f_t(w) = \tau^{(t)}$, and the main difference of this algorithm from the MWU scheme (Algorithm 1) is
845 that we use an additive/gradient-descent update, in lieu of the multiplicative update.

---

**Algorithm 4:** Gradient descent for spectral sample reweighing (Definition 2.1)

**Input:** A set of points $\{x_i\}_{i=1}^n$, an iteration count $T$, and step sizes $\eta_t$
**Output:** A point $\nu \in \mathbb{R}^d$ and weights $w \in \mathcal{W}_{n,\epsilon}$.

1 Let $w^{(1)} = \frac{1}{n}(1, 1, \cdots, 1)$.
2 **For** $t$ *from* 1 *to* $T$
3      Let $\nu^{(t)} = \sum_i w_i^{(t)} x_i$, $M^{(t)} = \sum_i w_i^{(t)} (x_i - \nu^{(t)})(x_i - \nu^{(t)})^T$.
4      Let $v^{(t)}$ be the top eigenvector of $M^{(t)}$ (with $\|v^{(t)}\| = 1$).
5      Compute $\tau_i^{(t)} = \langle v^{(t)}, x_i - \nu^{(t)} \rangle^2$.
6      Set $w_i \leftarrow w_i - \eta_t \tau^{(t)}$.
7      Project $w^{(t+1)}$ onto the set of good weights $\mathcal{W}_{n,\epsilon}$ (under $\ell_2$ distance).
8 **Return** $\nu^{(t^*)}, w^{(t^*)}$, where $t^* = \arg\min_t \|M^{(t)}\|$.

---

846 **Lemma H.2.** *Let $\rho$ be the squared diameter of the inputs points $\{x_i\}_{i=1}^n$. The cost function $f_t(\cdot)$ is*
847 $\sqrt{n}\rho$-*Lipschitz (in $\ell_2$ norm), for all t.*

848 *Proof.* Since $f_t$ is differentiable, we only need the bound $\|\nabla f_t\|$. We have that for all $t$ and $i$,

$$\tau_i^{(t)} = \langle v^{(t)}, x_i - \nu^{(t)} \rangle^2 \leq \|x_i - \nu^{(t)}\|_2^2 \leq \rho.$$

849 Therefore, $\|\nabla f_t\| = \|\tau^{(t)}\| \leq \sqrt{n}\rho$. $\qquad\square$

850 **Theorem H.3.** *Given $\{x_i\}_{i=1}^n$ and $\eta_t = R/L\sqrt{t}$ with $L = \sqrt{n}\rho, R = \sqrt{2}$, the online gradient*
851 *descent algorithm (based on Algorithm 4) yields a constant-factor approximation for the spectral*
852 *sample reweighing problem (Definition 2.1) in $O(nd^2/\epsilon^2)$ iterations and $O(n^2 d^3/\epsilon^2)$ total run-time.*

853 *Proof.* We first apply the PRUNE procedure of Lemma B.3 to bound the diameter. By Lemma B.2
854 and the guarantee of PRUNE, we can have $\rho = 16d\lambda/\epsilon$. Then we apply Algorithm 4.

855 We will use Lemma H.1 to analyze Algorithm 4. First, by Lemma H.2, we have $L = \sqrt{n}\rho$, and
856 further, since the $\ell_2$ diameter of the probability simplex can be (trivially) bounded by $\sqrt{2}$, $R = \sqrt{2}$.
857 Moreover, observe for any $t$,

$$f_t\left(w^{(t)}\right) = \left\langle w^{(t)}, \tau^{(t)} \right\rangle = \sum_i w_i^{(t)} \left\langle v^{(t)}, x_i - \nu^{(t)} \right\rangle^2 = v^{(t)T} M^{(t)} v^{(t)} = \left\| M^{(t)} \right\|_2.$$

Let $w \in \mathcal{W}_{n,\epsilon}$ be a weight that satisfies the spectral centrality condition. Then, from the regret guarantee (H.1),

$$\frac{1}{T}\sum_{t=1}^{T}\left\|M^{(t)}\right\|_{2} \leq \frac{1}{T}\sum_{t=1}^{T}\left\langle w, \tau^{(t)}\right\rangle + \frac{3LR}{2\sqrt{T}} \tag{H.2}$$

We bound the two terms on the right side individually.

(i) A bound on the first term follows exactly from the calculations we did in the analysis of MWU algorithm (Algorithm 1). In particular, from (B.12) we have

$$\frac{1}{T}\sum_{t=1}^{T}\langle w, \tau^{(t)}\rangle \leq 15\lambda + \frac{1}{3T}\sum_{t=1}^{T}\left\|M^{(t)}\right\|_{2}.$$

(ii) Observe that it suffices to set $T = 3L^2R^2/\lambda^2$ to bound the second term by $\lambda$.

Substituting the two bounds back into (H.2),

$$\frac{1}{T}\sum_{t=1}^{T}\left\|M^{(t)}\right\|_{2} \leq 16\lambda + \frac{1}{3T}\sum_{t=1}^{T}\left\|M^{(t)}\right\|_{2}. \tag{H.3}$$

Rearranging and dividing through immediately yields the desired guarantee.

Given that $L = \sqrt{n}\rho, R = \sqrt{2}$, we have that the iteration count $T = 6n\rho^2/\lambda^2$. Since $\rho = 16d\lambda/\epsilon$, $T = O(nd^2/\epsilon^2)$. For the run-time, note that instead of computing the exact largest eigenvector, we can use power method to find an $7/8$-approximate one. Observe that this suffices for our analysis of the method above. Finally, the Euclidean projection onto $\mathcal{W}_{n,\epsilon}$ can be computed in $O(n \log n)$ time [44]. This yields the desired run-time. □

## H.2 Extension to sub-gaussian setting

Theorem H.3 implies that a gradient descent-based algorithm (Algorithm 4) can be used for robust mean estimation under bounded covariance. We now extend the result to the sub-gaussian setting, showing that the same iteration and run-time complexity holds. The optimal estimation error we will aim for is $O(\epsilon\sqrt{\log(1/\epsilon)})$. We assume the spectral signature Lemma F.1 and the deterministic condition (F.1).

---

**Algorithm 5:** Gradient descent for sub-gaussian robust mean estimation

---

**Input:** A set of points $\{x_i\}_{i=1}^n$, step sizes $\eta_t$, an iteration count $T$, and parameter $\rho$
**Output:** A set of weights $w \in \mathcal{W}_{n,\epsilon}$.

---

1 Let $w^{(1)} = \frac{1}{n}\mathbb{1}_n$.
2 **For** $t$ *from* $1$ *to* $T$
3      Let $\nu^{(t)} = \sum_i w_i^{(t)}x_i$, $M^{(t)} = \sum_i w_i^{(t)}(x_i - \nu^{(t)})(x_i - \nu^{(t)})^T$.
4      Let $v^{(t)}$ be a $(1 - \epsilon^2)$-approximate largest eigenvector of $M^{(t)}$ (with $\|v^{(t)}\| = 1$). Fail with probability at most $\delta/T$.
5      **If** $\lambda^{(t)} = v^{(t)\top}M^{(t)}v^{(t)} \leq 1$, **return** $w^{(t)}$.
6      Compute $\tau_i^{(t)} = \left\langle v^{(t)}, x_i - \nu^{(t)}\right\rangle^2$.
7      Set $w_i \leftarrow w_i - \eta_t\tau^{(t)}$.
8      Project $w^{(t+1)}$ onto the set of good weights $\mathcal{W}_{n,\epsilon}$ (under $\ell_2$ distance).
9 **Return** $w^{(t^*)}$, where $t^* = \arg\min_t \|M^{(t)}\|$.

---

In particular, we will analyze Algorithm 5 and prove the following set of guarantees.

**Lemma H.4.** *Let $\epsilon$ be a sufficiently small constant and $\{x_i\}_{i=1}^n$ be $n$ points in $\mathbb{R}^d$. Assume the following (deterministic) conditions hold.*

880   (i) *There exists $\nu \in \mathbb{R}^d$ and $w \in \mathcal{W}_{n,\epsilon}$ such that*

$$\left\| \sum_{i=1}^{n} w_i \left( x_i - \nu \right) \left( x_i - \nu \right)^\top \right\| \leq 1 + O\left( \epsilon \log \left( 1/\epsilon \right) \right). \tag{H.4}$$

881   (ii) *If $\|M(w)\| \leq 1 + \lambda$, for some $\lambda \geq 0$, then for any $w \in \mathcal{W}_{n,\epsilon}$,*

$$\| \nu - \mu(w) \| \leq \frac{1}{1 - \epsilon} \left( \sqrt{\epsilon \lambda} + C\epsilon \sqrt{\log(1/\epsilon)} \right), \tag{H.5}$$

882   *Then, given $\{x_i\}_{i=1}^{n}$, a failure rate $\delta$ and $\rho$ such that $\rho \geq \tau_i^{(t)}$ for all $i$ and $t$, Algorithm 5 finds*
883   *$w' \in \mathcal{W}_{n,\epsilon}$ such that*

$$\|M(w')\| \leq 1 + O\left( \epsilon \log \left( 1/\epsilon \right) \right), \tag{H.6}$$

884   *with probability at least $1 - \delta$.*

885   *The algorithm terminates in $T = O(n\rho^2/\epsilon^2)$ iterations. Further, if $T = O(poly(n,d))$, then each*
886   *iteration takes $\widetilde{O}(nd \log \left( 1/\delta \right)/\epsilon^2)$ time.*

887   *Proof.* If the algorithm gets early stopped, then $\|M^{(t)}\| \leq 1 + O(\epsilon^2)$, so assumption (H.4) guarantees
888   that $\mu(w^{(t)})$ achieves the desired bound (H.6). We now assume that $\|M^{(t)}\| > 1$ for any $t$.

889   By the regret bound (Lemma H.1) and the inequality $\left\langle w^{(t)}, \tau^{(t)} \right\rangle \geq \left( 1 - \epsilon^2 \right) \left\| M^{(t)} \right\|_2$, for a $w$ that
890   satisfies assumption (H.4)

$$\frac{1 - \epsilon^2}{T} \sum_{t=1}^{T} \left\| M^{(t)} \right\|_2 \leq \frac{1}{T} \sum_{t=1}^{T} \left\langle w, \tau^{(t)} \right\rangle + \frac{3LR}{2\sqrt{T}}, \tag{H.7}$$

891   where $L = \sqrt{n}\rho$ and $R = \sqrt{2}$. For the first term, note that we may apply Claim F.3 and obtain

$$\frac{1}{T} \sum_{t=1}^{T} \left\langle w, \tau^{(t)} \right\rangle \leq 1 + O\left( \epsilon \log(1/\epsilon) \right) + \frac{2\epsilon}{(1 - \epsilon)^2} \frac{1}{T} \sum_{t=1}^{T} \left\| M^{(t)} \right\| - \frac{2\epsilon}{(1 - \epsilon)^2}$$

892   By setting $T = 3L^2 R^2/\epsilon^2 = O(n\rho^2/\epsilon^2)$, we can bound the second term by $O(\epsilon)$

893   Substituting the bounds back into (H.7), we obtain

$$\frac{1 - \epsilon^2}{T} \sum_{t=1}^{T} \left\| M^{(t)} \right\|_2 \leq 1 - \frac{2\epsilon}{(1 - \epsilon)^2} + O(\epsilon \log(1/\epsilon)) + \frac{1}{T} \sum_{t=1}^{T} \frac{2\epsilon}{(1 - \epsilon)^2} \left\| M^{(t)} \right\|$$

894   For sufficiently small $\epsilon$, we can move the last term to the left side and divide through. This immedi-
895   ately yields that

$$\frac{1}{T} \sum_{t=1}^{T} \left\| M^{(t)} \right\|_2 \leq 1 + O(\epsilon \log(1/\epsilon)).$$

896   The run-time follows from the cost of computing $(1 - \epsilon^2)$-approximate largest eigenvector via power
897   iteration. □

898   Using the same argument for Theorem F.4, Lemma H.4 implies the following theorem.

899   **Theorem H.5.** *Let $S = \{x_i\}_{i=1}^{n}$ be an $\epsilon$-corrupted set of $n$ samples from a sub-gaussian distribution*
900   *over $\mathbb{R}^d$, with mean $\mu$ and identity covariance. Suppose $n \geq \widetilde{\Omega}(d/\epsilon^2)$. Then given $S$, there is an*
901   *algorithm (based on Algorithm 5) that finds $\widehat{\mu}$ such that with high constant probability $\|\widehat{\mu} - \mu\| \leq$*
902   *$O\left( \epsilon \sqrt{\log(1/\epsilon)} \right)$.*

903   *The algorithm runs in $\widetilde{O}(nd^2/\epsilon^2)$ iterations and $\widetilde{O}(n^2 d^3/\epsilon^2)$ total time.*

## H.3 Equivalence with [9]

The recent work of Cheng, Diakonikolas, Ge and Soltanolkotabi [9] studies a gradient-descent-based algorithm for solving the following non-convex formulation of robust mean estimation.

$$\min \ \|\Sigma_w\| \quad \text{such that } w \in \mathcal{W}_{n,\epsilon}.$$

where $\Sigma_w = \sum_{i=1}^n w_i(x_i - \mu(w))(x - \mu(w))^\top$. This is equivalent to

$$\min_w \ \max_{u \in \mathbb{S}^{d-1}} \ F(w, u) = u^\top \Sigma_w u \quad \text{such that } w \in \mathcal{W}_{n,\epsilon}.$$

The sub-gradient of $F(w, u)$ with respect to $w$ (for a fixed $u$) is given by

$$\nabla_w F(w, u) = Xu \odot Xu - 2\left(w^\top Xu\right) Xu, \tag{H.8}$$

where $X \in \mathbb{R}^{n \times d}$ is the data matrix whose the $i$th row is $x_i$.

Based on the observation, they consider and analyze an algorithm that computes a (approximately) maximizing $u$ and performs a projected gradient descent on $w$ each iteration.

Since Algorithm 4 can be directly applied to the same robust setting (Corollary C.4), it is natural to consider the relationships between the two algorithms. Indeed, one can argue that they are essentially the same. First, we unpack our gradient update (*i.e.*, the spectral scores) of iteration $t$. Note that

$$
\begin{aligned}
\nabla_i f_t(w^{(t)}) = \tau_i^{(t)} &= \left\langle v^{(t)}, x_i - \nu^{(t)} \right\rangle^2 \\
&= \left\langle v^{(t)}, x_i \right\rangle^2 + \left\langle v^{(t)}, \nu^{(t)} \right\rangle^2 - 2 \left\langle v^{(t)}, x_i \right\rangle \left\langle v^{(t)}, \nu^{(t)} \right\rangle \\
&= \left( Xv^{(t)} \odot Xv^{(t)} \right)_i + \left( w^{(t)\top} Xv^{(t)} \right)^2 - 2 \left( w^{(t)\top} Xv^{(t)} \right) \left( Xv^{(t)} \right)_i
\end{aligned}
$$

since $\nu^{(t)} = \sum_i w_i^{(t)} x_i = X^T w^{(t)}$, where $\odot$ denotes entrywise product of vectors. Let $C_t = w^{(t)\top} Xv^{(t)}$. Therefore, we can rewrite the gradient as

$$\nabla f_t(w^{(t)}) = C_t^2 \cdot \mathbb{1}_n + Xv^{(t)} \odot Xv^{(t)} - 2C_t \cdot Xv^{(t)}$$

Note that the gradient (H.8) used in [9] is exactly the same as above, except without the term of all-one vector $C_t^2 \cdot \mathbb{1}_n$. In the gradient update step, the additional term reduces the weight of every point uniformly by the same quantity $C_t^2$. However, observe that by Pythagorean theorem, the (Euclidean) projection onto $\mathcal{W}_{n,\epsilon}$ can be decomposed into two (sequential) steps: (1) first an orthogonal projection onto the affine subspace containing $\mathcal{W}_{n,\epsilon}$, and then (2) a projection onto $\mathcal{W}_{n,\epsilon}$ itself. Note that reducing each coordinate by the same quantity or not results in the same vector by the first step. Therefore, the two algorithms yield the same sequence of iterates $(w^{(t)})_t$.

## Footnotes

[2]The $1/\epsilon$ dependence in the run-time can be removed by a simple bucketing trick due to [12]; also see