[Reviews · NeurIPS 2020]

Review 1

Summary and Contributions: The paper considers the problem of mean estimation when the data is heavy-tailed or contaminated with outliers. The authors show a key result connecting the two seemingly disparate notions of robustness. In particular, the authors show an equivalence between two different notions of centrality (1) combinatorial center of Lugosi-Mendelson which is used in Heavy-Tailed Estimation and (2) the spectral center which is used in Outlier-Robust estimation. With the use of this equivalence, the authors show that the algorithms derived in one field can be used in the other, and in particular, show that the filtering based algorithms, which approximate the spectral center can be used for heavy-tailed mean estimation.

Strengths: I think this a strong paper. In my opinion, the equivalence between the two notions is an interesting result, and I hope to see future work building on this result.

Weaknesses: Nothing as such. ### After Rebuttal #### I thank the reviewers for their thoughtful response. After looking at it and reading the other reviews, my score remains the same.

Correctness: Yes.

Clarity: Yes.

Relation to Prior Work: Yes

Reproducibility: Yes

Additional Feedback:


Review 2

Summary and Contributions: This paper connects two important settings of high-dimensional mean estimation: 1) the robust setting where an epsilon-fraction of the examples are adversarially corrupted, and 2) the heavy-tailed setting where the data is only guaranteed to have bounded covariance. For both settings, many breakthroughs happened only recently, but this paper matches the SOTA of both problems by reducing them to a single deterministic problem, which the paper solves using previous techniques with simplified analyses. The paper shows an insightful lemma (equivalence between two notions of center) when reducing the heavy-tailed setting to the deterministic setting.

Strengths: - The theory developed in this paper is clean and powerful. There are many previous algorithms for robust mean estimation and their analyses all appear somewhat complicated while sharing some implicit commonality. This paper unifies these previous results into a single framework that will likely assist in future research. - The paper provides a new viewpoint for the heavy-tailed mean estimation problem, revealing fundamental nature of the problem. - The paper would be of great interest to people who study robust statistics and robust machine learning, and to all statisticians in general.

Weaknesses: - There is no doubt that most results in the paper regarding robust/heavy-tailed mean estimation, when viewed in isolation, are already known or implicitly known. However, the paper gives simplified proofs, improved analyses, and new insights, unifying all these previous results into a single clean framework.

Correctness: I do not see any technical errors in the paper.

Clarity: The paper is extremely well written, presenting so many results in a unified and structured manner.

Relation to Prior Work: The paper unifies, simplifies, and improves many previous results. As far as I am aware, related previous works are clearly discussed.

Reproducibility: Yes

Additional Feedback: - I do not see a good reason why the equivalence between the two notions of center is called "duality". It would help me if the authors could explain this. - Line 294: "to to" -> "to" ========================= I thank the authors for the response. Since the two notions of center are of different nature (spectral VS combinatorial), the motivation for calling the equivalence between them as duality is still a bit unclear to me. This, however, does not affect my overall evaluation of the paper, and I look forward to seeing the paper on the acceptance list.


Review 3

Summary and Contributions: The paper considers the problem of robust mean estimation under the \epsilon-corruption model and heavy-tailed distributions in a unified framework. It proposes a meta-problem (spectral sample reweighing) and a multiplicative weight meta-algorithm that is proved to solve the problem in appropriate runtime via a standard regret analysis. It then shows mean estimation under the \epsilon-corruption model can be cast into an instance of the meta-problem and, through a duality result for spectral and combinatorial centers, shows further that mean estimation under heavy-tailed distribution can also be seen as a special case of the meta-problem. Therefore, the proposed meta-algorithm can solve both mean estimation problems (with some adaptation under heavy-tailed distributions).

Strengths: The paper considers mean estimation under \epsilon-corruption model and heavy-tailed distributions, which is a central topic in robust learning and statistics. The paper is very clearly written and core concepts are well explained. The meta-framework it proposes is simple and streamlined, and connects robust mean estimation with online learning. The equivalence results established in the paper between spectral and combinatorial centers is very interesting. Overall, the proposed meta-framework and the equivalence results can help us better understand the problem of robust mean estimation.

Weaknesses: In showing the equivalence between spectral and combinatorial centers (Prop 3.1 and 3.2), the authors require \epsilon to be some constants (0.3 and 0.1 respectively). The constants seem quite arbitrary and not very intuitive. It would be great if the equivalence relationship can be shown to hold for any small enough \epsilon. Nevertheless, the current form of the equivalence results is already interesting and valuable.

Correctness: The arguments appear correct.

Clarity: Yes.

Relation to Prior Work: Yes.

Reproducibility: Yes

Additional Feedback: - L153, [46] work -> [46] - L263, missed a period at the end of the last sentence ---------------- Rebuttal ---------------- I have read the rebuttal and it has addressed my concerns.

[Author Response · NeurIPS 2020]

We thank all the reviewers for their thoughtful comments and suggestions. We will fix all the typos and mechanical errors in the camera-ready and future arXiv version. Reviewer-specific comments follow.

**Reviewer #3.** Regarding our equivalence theorem between heavy-tailed and robust mean estimator, we use the word "duality" here only to highlight that the arguments crucially leverage the strong duality of SDP. We will clarify this point explicitly in the full version.

**Reviewer #4.** The constants in Proposition 3.1 and 3.2 are immaterial. Indeed, they are set up this way to make the argument in the later sections to go through. For all the applications, the current form suffices. If needed, one could tweak our proof to obtain the same type of results with different constants. We will make a clarification in the full version.

[Meta-Review · NeurIPS 2020]

This paper makes a connection between outlier-robust mean estimation and mean estimation with subgaussian rates for heavy tailed distributions and show an equivalence between natural estimators for both these estimation tasks. As a consequence, one derives the algorithms for outlier-robust mean estimation can be used to solve the heavy-tailed mean estimation problem. The reviewers found this work interesting in its algorithmic contributions and insights on two natural tasks in robust estimation. I recommend acceptance of this paper to the NeurIPS program.